# 4-hexylresorcinol-induced protein expression changes in human umbilical cord vein endothelial cells as determined by immunoprecipitation high-performance liquid chromatography

**Yeon Sook Kim[1], Dae Won Kim[2], Seong-Gon Kim[3]⊕\*, Suk Keun Lee[4]⊕\***

**1** Department of Dental Hygiene, College of Health & Medical Sciences, Cheongju University, Cheongju, South Korea, **2** Department of Oral Biochemistry, College of Dentistry, Gangneung-Wonju National University, Gangneung, Korea, **3** Department of Oral and Maxillofacial Surgery, College of Dentistry, Gangneung-Wonju National University, Gangneung, South Korea, **4** Department of Oral Pathology, College of Dentistry, Gangneung-Wonju National University, and Institute of Hydrogen Magnetic Reaction Gene Regulation, Gangneung, South Korea

⊕ These authors contributed equally to this work.
\* sukkeunlee2@naver.com (SKL); kimsg@gwnu.ac.kr (SGK)

**Data Availability Statement:** All relevant data are within the manuscript and its Supporting Information files.

## Abstract

4-Hexylresorcinol (4HR) is used as a food preservative and an ingredient of toothpaste and cosmetics. The present study was performed using 233 antisera to determine the changes in protein expression induced by 4HR in human umbilical cord vein endothelial cells (HUVECs), and evaluated the 4HR-induced effects in comparison with previous results (Kim et al., 2019). Similar to RAW 264.7 cells, 4HR-treated HUVECs showed decreases in the expression of the proliferation-related proteins, cMyc/MAX/MAD network proteins, p53/RB and Wnt/β-catenin signaling, and they showed inactivation of DNA transcription and protein translation compared to the untreated controls. 4HR upregulated growth factors (TGF-β1, β2, β3, SMAD2/3, SMAD4, HGF-α, Met, IGF-1) and RAS signaling proteins (RAF-B, p38, p-p38, p-ERK-1, and Rab-1), and induced stronger expression of the cellular protection-, survival-, and differentiation-related proteins in HUVECs than in RAW 264.7 cells. 4HR suppressed NFkB signaling in a manner that suggests potential anti-inflammatory and wound healing effects by reducing M1 macrophage polarization and increasing M2 macrophage polarization in both cells. 4HR-treated HUVECs tended to increase the ER stress mediators by upregulating eIF2AK3, ATF4, ATF6, lysozyme, and LC3 and downregulating eIF2α and GADD153 (CHOP), resulting in PARP-1/AIF-mediated apoptosis. These results indicate that 4HR has similar effects on the protein expression of HUVECs and RAW 264.7 cells, but their protein expression levels differ according to cell types. The 4HR-treated cells showed global protein expression characteristic of anticancer and wound healing effects, which could be alleviated simultaneously by other proteins exerting opposite functions. These results suggest that although 4HR has similar effects on the global protein expression of HUVECs and RAW 264.7 cells, the 4HR-induced molecular interferences in those cells are complex enough to produce variable protein expression, leading different cell functions.

**Funding:** Unfunded studies.

**Competing interests:** NO authors have competing interests.

Moreover, HUVECs have stronger wound healing potential to overcome the impact induced by 4HR than RAW 264.7 cells.

## Introduction

Resorcinolic lipids were suggested to induce dormancy in micro-organisms, and they have an anti-microbial effect [1]. 4-Hexylresorcinol (4HR) is a synthetic resorcinolic lipid that is synthesized from resorcinol and caproic acid [2]. 4HR has a long alkyl group, and can bind to the hydrophobic pocket of enzymes such as tyrosinase [3]. 4HR has been used as a food preservative because of its strong inhibitory effect [4]. The antineoplastic effect of 4HR is derived from its growth inhibition and the resulting apoptosis of cancer cells [5]. Interestingly, the two hydroxyl groups in 4HR have antioxidant activity and are associated with enzymes [6], such as glutathione peroxidase and glutathione reductase [7].

Under a two-year gavage study conducted by administering 0, 62.5 or 125 mg/kg (0, 62.5 or 125 μg/g) to groups of 50 F344/N rats and 50 B6C3F1 mice of each sex, five days per week, there was no significant differences in survival and no evidence of carcinogenic activity [8]. The oral $LD_{50}$ of 4HR was 550 mg/kg body weight in rat [9, 10], 475 mg/kg in Guinea-pig [11], approximately 750 mg/kg in rabbit [11], and 200–1000 mg/kg in mice (subcutaneous injection with 5% 4HR in olive oil; 750–1000 mg/kg, intraperitoneal injection with 5% 4HR in olive oil; 200 mg/kg, with 1% 4HR aqueous emulsion; 300 mg/kg) [10]. The probable oral $LD_{50}$ of 4-hexylresorcinol in humans has been estimated to be between 500 and 5000 mg/kg [12]. These data indicate 4 HR may have relatively wide range of applicable dose in animals and human, and that the dose used in this study, 10 μg/mL, is within a safe range and might be free of toxic chemical hazard.

4HR is excreted via the urine mainly in the form of an ethereal sulfate conjugate [13]. In the animal experiments [14], dogs were given single doses of 1 or 3 g 4HR (equivalent to 100 or 300 mg/kg body weight) as crystals in gelatin capsules or as a solution in olive oil, and its excretion in urine and feces was monitored. After administrating 1 g crystalline compound, 29% of the dose was detected in urine and 67% in feces. When the dose was increased to 3 g, 17% and 73% was excreted in urine and feces, respectively. Urinary excretion was rapid, mainly in the first 6 h, and levels were virtually undetectable 12 h after the lower dose and 24–36 h following the higher dose. When 4HR was administered in olive oil, a dose of 1 g resulted in 17% and 76% was excreted in urine and feces, respectively, while 3 g, 10% and 80% was excreted in urine and feces, respectively. When two men received doses of 1 g 4HR, an average of 18% of the dose was recovered in urine within the first 12 h. Thereafter, the compound was not detected in urine samples. Fecal excretion accounted for 64% of the dose [15]. These results suggest the metabolic degradation of 4HR is vigorous for 6 h and persists until 24 h. Therefore, the present study performed 4HR treatment for 24 h in cell culture experiment.

The intracellular concentration of reactive oxygen species (ROS) in macrophages is closely associated with foreign body reactions [16]. Indeed, 4HR-incorporated biomaterials inhibits the formation of foreign body giant cells [17], but produces rich vascularity [17, 18]. The administration of 4HR increased the expression of vascular endothelial growth factor (VEGF) via hypoxia-inducible factor (HIF)-independent pathway in macrophages, RAW 264.7 cells [19, 20]. 4HR also increases the expression of different angiogenesis factors [21, 22] and proteins associated with M2 macrophage polarization in RAW264.7 cells [19, 20].

As the 4HR-induced genome-wide protein interactions have not been clarified, the present study examined 4HR-induced protein expression in human umbilical cord vein endothelial

cells (HUVECs) by immunoprecipitation high-performance liquid chromatography (IP-HPLC) with the previous results in RAW 264.7 cells used for comparison [19]. IP-HPLC has been developed to quantify peptides or organic chemicals [23, 24]. This study used the advanced IP-HPLC protocol for an analysis of the protein expression levels. This IP-HPLC protocol has been designed for the analysis of exudates from inflammatory tissues [25–28], protein expression profile in the tumor samples [29], and protein expression changes after the administration of coffee extract in cells [30] and animal [31]. IP-HPLC is based on immuno-precipitation after an objective antigen-antibody reaction in a column filled with protein A/G agarose beads. Using precision UV spectroscopy, the ratio of the protein level can be deter-mined by comparing it with a control protein level. Although the basic principle is similar to that of an enzyme-linked immunosorbent assay, IP-HPLC is a rapid, accurate, and reproduc-ible method [25–27, 32].

In the previous IP-HPLC-based study [19], the administration of 4HR to macrophages (RAW 264.7 cells) produced significant reductions in the expression of proliferation-, inflam-mation-, protection-, and oncogenesis-related proteins but uncertain effects on angiogenesis and osteogenesis. As angiogenesis and osteogenesis are important phenomena for tissue organogenesis, it is suggested that HUVECs (endothelial cells), which are the counterparts of RAW 264.7 cells (macrophages) in inflammation and wound healing, also need to be explored for their global protein expression changes induced by 4HR. Therefore, in the present study, HUVECs were used to investigate the 4HR-induced protein expression changes by IP-HPLC, and results were compared with a previous report [19].

## Materials and methods

### HUVEC culture in the presence of 4HR

HUVECs were purchased from Lonza (Walkersville, MD USA). The culture medium for the HUVECs was an endothelial basal medium contained hydrocortisone (1 µg/mL), bovine brain extract (12 µg/mL), gentamicin (50 µg/mL), amphotericin-B (50 ng/mL), epidermal growth factor (10 ng/mL), vascular endothelial growth factor (1 ng/mL), basic human fibroblast growth factor (10 ng/mL), heparin (22.5 µg/mL), ascorbic acid (1 µg/mL), and 10% fetal calf serum (EGM-2; Clonetics, Lonza, Walkersville, MD USA). HUVECs were grown in a 5% $CO_2$ incubator at 37.5˚C. The cells were also tested for mycoplasma regularly to confirm that only mycoplasma-free cells were assayed.

When 70–80% confluent HUVECs were spread on Petri dish surfaces, cells were treated with 10 µg/mL 4HR (Sigma-Aldrich, St. Louis, MO, USA) for 8, 16, or 24 h, while control cells were treated with 100 µL of normal saline. If 20% of 4HR administered orally was absorbed in different animals, the cellular level dose of 4HR was calculated to be 20 or 60 mg/kg in dog [14], 12 mg/kg in cat [33], 12.5 or 25 mg/kg in rat [8], and 25, 50 or 100 mg/kg in mice [8]. 4HR is commonly employed in 1:1000 solution or glycerite (0.01%, 194 µg/mL) in mouthwashes or pharyngeal antiseptic preparation [34]. Therefore, the dose used in the present study, 10 µg/mL, is safe within the experimental range, and the previous studies used the same dose of 4HR had showed characteristic protein expression in cell culture [20, 21, 35–37]. Cultured cells were harvested with protein lysis buffer (PRO-PREP[TM], iNtRON Biotechnology, Daejeon, Korea) in ice, and immediately preserved at -70˚C until required.

### Direct cell counting assay for the proliferation index

HUVECs were cultured on the surfaces of two-well culture slide dishes (SPL, Korea) until they reached 50% confluence, and were then treated with 4HR at 10 µM for 8, 16, or 24 h. The

control was treated with normal saline only. The cells on the culture slides were fixed with a 10% buffered formalin solution, stained with hematoxylin, and observed by optical microscope (CX43, Olympus, Japan) at x200 magnification. Thirty representative images were digitally captured in each group (DP-73, Olympus Co., Japan), followed by a cell counting assay using the IMT i-solution program (version 21.1; Martin Microscope, Vancouver, Canada). The results were plotted on a graph.

## Immunocytochemical analysis

When approximately 70% confluent HUVECs were spread over the surfaces of two-well culture slide dishes, the cells were treated with 10 μg/mL 4HR for 8, 16, or 24 h, while the control cells were treated with 100 μL of normal saline. The cells on the culture slides were pretreated with 70% ethanol for 30 min, fixed with 10% buffered formalin solution, and applied for immunohistochemistry using the antisera of E-cadherin, VE-cadherin, TGF-β1, caspase 3 (a polyclonal antibody (PoAb) raised against amino acids 1–277 representing full length procaspase-3 of human origin), PARP-1 (a PoAb raised against amino acids 764–1014 mapping at the C-terminus of PARP-1 of human origin), lysozyme, PERK, eIF2α, ATF4, GADD153 (CHOP), and LC3 (the same antibodies used in IP-HPLC). Immunocytochemical (ICC) staining was performed using the indirect triple sandwich method on the Vectastatin system (Vector Laboratories, USA), and visualized using a 3-amino-9-ethylcarbazole solution (Santa Cruz Biotechnology, USA). The results were observed by optical microscope, and their characteristic images were captured (DP-73, Olympus Co., Japan) and illustrated.

## Western blot analysis

The selected protein expression levels of E-cadherin, VE-cadherin, TGF-β1, LC3, PERK, eIF2α, ATF4, GADD153, PARP-1, c-PARP-1 (using a PoAb raised against a short amino acid sequence containing the neoepitope at Gly 215 of PARP of human origin), c-caspase 3 (using a PoAb raised against a synthetic peptide corresponding to amino-terminal residues adjacent to (Asp175) in human caspase-3), and AIF for the HUVECs treated with 10 μg/mL 4HR for 8, 16, or 24 h were examined by western blot. The control was treated with normal saline only. The cells were collected with phosphate-buffered saline (PBS), treated with trypsin-ethylene-diamine-tetra-acetic acid (trypsin-EDTA) for one minute, and washed with PBS, and followed by cell lysis with ice-cold RIPA buffer (Sigma Aldrich, USA). The lysates were centrifuged at 12,000 g for 20 min at 4° C. The protein concentration of the supernatant was quantified using a Bradford assay (BioRad, USA). Equal amounts (30 μg/lane) of the sample proteins were separated by 8, 10, 15, or 20% sodium dodecyl sulfate-polyacrylamide gel electrophoresis (SDS-PAGE) in Tris-glycine SDS running buffer (25 mM Tris, 0.1% SDS, and 0.2M glycine) to analyze the target proteins with the protein marker. After the proteins were transferred from the gel to a nitrocellulose membrane, the membranes were blocked with 5% nonfat dry milk in TBST buffer (25 mM Tris-HCl, 150 mM NaCl, 0.1% Tween 20, pH 7.5) for 1 h. After washing three times with TBST buffer, the membrane was incubated with each primary antibody (dilution ratio = 1:1000, the same antibody used in IP-HPLC) and horseradish peroxidase-conjugated secondary antibody for 1 h separately. The protein bands were then detected using an enhanced chemiluminescence system (Amersham Pharmacia Biotech, Piscataway, NJ, USA) according to the manufacturer's instructions and digitally imaged using a ChemiDoc XRS system (Bio-Rad Laboratories, Hercules, CA, USA). The level of β-actin expression was used as an internal control to normalize the expression of the target proteins.

## Immunoprecipitation High-Performance Liquid Chromatography (IP-HPLC)

The protein A/G agarose bead column (Amicogen, Jinju, Korea) was separately pre-incubated with 1 μg of 233 different antisera shown in Table 1 in Tris-HCl buffer (pH 7.5, 0.1% Tween

**Table 1. Antibodies used in the study.**

| Proteins | No. | Antibodies |
|---|---|---|
| Proliferation-related proteins | 11 | Ki-67*, PCNA*, CDK4*, MPM2*, PLK4*, cyclin D2, p14*, p15/16*, p21*, p27*, lamin A/C* |
| cMyc/MAX/MAD network | 4 (1) | cMyc*, MAX*, MAD-1*, (p27) |
| p53/Rb/E2F signaling | 4 (1) | p53, Rb-1#, E2F-1*, (CDK4) |
| Wnt/β-catenin signaling | 7 | Wnt1*, β-catenin*, APC*, snail*, TCF-1*, E-cadherin*, VE-cadherin& |
| Epigenetic modification | 7 | histone H1*, DMAP1*, KDM4D$, HDAC10$, MBD4*, DNMT1*, PCAF* |
| Protein translation | 7 | DOHH!, DHS!, eIF5A-1!, eIF2AK3 (PERK)*, p-eIF2AK3 (Thr981)*, eIF2α^, p-eIF2α (Ser51)^ |
| Growth factor | 20 | TGF-β1#, TGF-β2*, TGF-β3*, SMAD2/3, SMAD4*, p-SMAD4, HGFα*, Met*, FGF-1*, FGF-2*, FGF-7*, GH*, GHRH*, IGF-1*, IGFIIR`, PDGF-A*, CTGF*, HER1*, HER2*, ERβ*, |
| RAS signaling proteins | 21 | NRAS$, KRAS$, HRAS, STAT3*, PI3K, RAF-B*, JNK-1*, p-JNK-1, JAK2$, ERK-1*, p-ERK-1 (T202, Y204)$, Rab 1*, p38, p-p38 (Thr180, Tyr182), pAKT1/2/3 (Thr308), AKAP, mTOR@, PTEN, PKC*, p-PKC1α (Thr514)@, SOS-1/2* |
| NFkB signaling proteins | 19 (11) | NFkB*, IKK*, (p38*, p-p38*), GADD45*, GADD153* (CHOP)*, (mTOR@, PKC*, p-PKC1α@), NRF-2*, (JAK2, ERK-1*, p-ERK-1$), PGC-1α, (pAKT1/2/3, AKAP, SRC-1*), MDR, AMPKα@, |
| Upregulated inflammatory proteins | 20 | IL-1*, IL-10*, IL-12*, cathepsin K*, lysozyme*, granzyme B, lactoferrin, M-CSF, β-defensin-1, β-defensin-2, β-defensin-3, CD28, Pdcd-1/1 (CD279), PECAM-1 (CD31)*, HCAM (CD44)*, ICAM-1 (CD54)*, versican*, COX1*, COX2*, kininogen* |
| Downregulated inflammatory proteins | 25 | TNFα@, IL-6*, IL-8, IL-28*, LTA4H&, CXCR4*, MMP-1, -2, -3$, -9$, -10, -12$, cathepsin C*, cathepsin G*, MCP-1, LL-37, α1- antitrypsin &, CD20, CD34, CD68, CD80 (B7-1), CD99 (MIC2), NCAM (CD56), VCAM-1 (CD106), CTLA4*, TLR3* |
| p53-mediated apoptosis | 15 (1) | (p53*), MDM2*, BAD*, BID*, BAK*, NOXA*, PUMA*, BAX*, BCL2*, APAF-1*, caspase 9*, c-caspase 9*, PARP-1*, c-PARP-1*, AIF* |
| FAS-mediated apoptosis | 9 (1) | FASL*, FAS*, FADD*, FLIP*, (BID*), c-caspase 8*, c-caspase 10*, caspase 3*, c-caspase 3* |
| Protection- and survival-related proteins | 20 (4) | PKC, p-PKC1α, (pAKT1/2/3), HSP-27*, HSP-70*, HSP-90*, LC3, (AMPKα, PGC-1α), TERT*, SP-1@, SP-3@, NOS-1$, leptin, PLC-β2, HO-1*, SOD-1*, GSTO1*, SVCT2&, (NRF2) |
| Differentiation-related proteins | 18 (1) | α-actin*, p63$, vimentin*, TGase-2, -4, caveolin-1*, GLI1, Jagged1*, Notch1, S-100*, AP1M1*, CaM*, cystatin A*, SHH*, FAK*, (PLC-β2), integrin α5*, CRIP-1* |
| Endoplasmic reticulum stress-related proteins | 11 (10) | (HSP-27, HSP-70, eIF2AK3* (PERK), p-EIF2AK3 (p-PERK)), ATF4*, ATF6*, (GADD153 (CHOP)*, LC3, AIF, AP1M1, endothelin-1, PGC-1α) |
| Oncogenesis-related proteins | 17 (2) | PTEN&, BRCA1&, BRCA2&, NF-1*, (MBD4), ATM*, PTCH-1, maspin*, DMBT1*, PIM-1*, CEA$, 14-3-3*, survivin@, mucin 1*, mucin 4*, YAP*, (CRIP1) |
| Angiogenesis-related proteins | 23 (9) | HIF-1α&, angiogenin$, VEGF-A*, VEGF-C*, VEGFR2*, p-VEGFR2 (Y951), vWF$, CMG2$, FLT-4$, LYVE-1*, (FGF-2, PDGF-A, MMP-2, MMP-10), endothelin-1*, plasminogen*, PAI-1*, fibrinogen*, (kininogen-1, HCAM*, VCAM-1*, ICAM-1*, PECAM-1*) |
| Osteogenesis-related proteins | 17 (5) | BMP-2*, BMP-3*, BMP-4*, OPG*, RANKL*, osteocalcin*, osteopontin*, osteonectin*, RUNX2*, osterix*, (TGF-β1), ALP*, (cathepsin K*, HSP-90*, versican*), aggrecan*, (CTGF*) |

(*Continued*)

**Table 1.** (Continued)

| Proteins | No. | Antibodies |
|---|---|---|
| Control housekeeping proteins | 3 | α-tubulin*, β-actin*, GAPDH* |
| Total | 278 (46) | |

* Santa Cruz Biotechnology, USA

# DAKO, Denmark

$ Neomarkers, CA, USA

@ ZYMED, CA, USA

& Abcam, Cambridge, UK

^Cell signaling technology, USA

! kindly donated from Dr. M. H. Park in NIH, USA [38], the number of antibodies overlapped; ().

**Abbreviations:** AIF; apoptosis inducing factor, AKAP; A-kinase anchoring proteins, ALP; alkaline phosphatase, AMPK; AMP-activated protein kinase, pAKT; v-akt murine thymoma viral oncogene homolog, p-Akt1/2/3 phosphorylated (p-Akt, Thr 308), APAF-1; apoptotic protease-activating factor 1, APC; adenomatous polyposis coli, ATF4; activating transcription factor 4, ATM; ataxia telangiectasia caused by mutations, BAD; BCL2 associated death promoter, BAK; BCL2 antagonist/killer, BAX; BCL2 associated X, BCL2; B-cell leukemia/lymphoma-2, BID; BH3 interacting-domain death agonist, BMP-2; bone morphogenesis protein 2, BRCA1; breast cancer type 1 susceptibility protein, c-caspase 3; cleaved-caspase 3, CaM; calmodulin, CD3; cluster of differentiation 3, CDK4; cyclin dependent kinase 4, CEA; carcinoembryonic antigen, CHOP (GADD153); C/EBP homologous protein, CMG2: capillary morphogenesis protein 2, COX-1; cyclooxygenase-1, CTGF connective tissue growth factor, CTLA4; cytotoxic T lymphocyte-associated protein-4, CXCR4; C-X-C chemokine receptor type 4, DHS; deoxyhypusine synthase, DMAP1; DNA methyltransferase 1 associated protein, DMBT1; deleted in malignant brain tumors 1, DNMT1; DNA 5-cytosine methyltransferase 1, DOHH; deoxyhypusine hydroxylase, E2F-1; transcription factor, eIF2AK3 (PERK); protein kinase R (PKR)-like endoplasmic reticulum kinase (PERK), eIF5A-1; eukaryotic translation initiation factor 5A-1, ERK-1; extracellular signal-regulated protein kinase 1, ERβ; estrogen receptor beta, FADD; FAS associated via death domain, FAK; focal adhesion kinase, FAS; CD95/Apo1, FASL; FAS ligand, FGF-1; fibroblast growth factor-1, FLIP; FLICE-like inhibitory protein, FLT-4; Fms-related tyrosine kinase 4, GADD45; growth arrest and DNA-damage-inducible 45, GAPDH; glyceraldehyde 3-phosphate dehydrogenase, GH; growth hormone, GHRH; growth hormone-releasing hormone, GSTO1; glutathione S-transferase ω 1, HCAM (CD44); homing cell adhesion molecule, HDAC10; histone deacetylase 10, HER1; human epidermal growth factor receptor 1, HGFα; hepatocyte growth factor α, HIF-1α: hypoxia inducible factor-1α, HO-1; heme oxygenase 1, HRAS; GTPase HRas, HSP-70; heat shock protein-70, ICAM-1 (CD54); intercellular adhesion molecule 1, IGF-1; insulin-like growth factor 1, IGFIIR; insulin-like growth factor 2 receptor, IKK; ikappaB kinase, IL-1; interleukin-1, JAK2; Janus kinase 2, JNK-1; Jun N-terminal protein kinase, KDM4D; Lysine-specific demethylase 4D, KRAS; V-Ki-ras2 Kirsten rat sarcoma viral oncogene homolog, LC3; microtubule-associated protein 1A/1B-light chain 3, LTA4H; leukotriene A4 hydrolase, LYVE-1: lymphatic vessel endothelial hyaluronan receptor 1, MAD-1; mitotic arrest deficient 1, MAX; myc-associated factor X, MBD4; methyl-CpG-binding domain protein 4, MCP-1; monocyte chemotactic protein 1, M-CSF; macrophage colony-stimulating factor, MDM2; mouse double minute 2 homolog, MDR; multiple drug resistance, MMP-1; matrix metalloprotease-1, MPM2; mitosis phase promoting factor (MPF) recognized by a mitosis-specific monoclonal antibody, mitotic protein monoclonal 2, mTOR; mammalian target of rapamycin, cMyc; V-myc myelocytomatosis viral oncogene homolog, NFkB; nuclear factor kappa-light-chain-enhancer of activated B cells, NCAM (CD56); neural cell adhesion molecule 1, NF-1; neurofibromin 1, NFkB; nuclear factor kappa-light-chain-enhancer of activated B cells, NOS-1; nitric oxide synthase 1, NOXA; Phorbol-12-myristate-13-acetate-induced protein 1, NRAS; neuroblastoma RAS Viral Oncogene homolog, NRF2; nuclear factor (erythroid-derived)-like 2, OPG; osteoprotegerin, PAI-1; plasminogen activator inhibitor-1, PARP-1; poly-ADP ribose polymerase 1, c-PARP-1; cleaved-PARP-1, PCAF; p300/CBP-associated factor, PCNA; proliferating cell nuclear antigen, Pdcd-1/1 (CD279); programmed cell death protein 1, PDGF-A: platelet-derived growth factor-A, PECAM-1 (CD31); platelet endothelial cell adhesion molecule-1, PERK; protein-like endoplasmic reticulum kinase, PGC-1α; peroxisome proliferator-activated receptor gamma coactivator 1-α, PI3K; phosphatidylinositol-3-kinase, PIM-1; Proto-oncogene serine/threonine-protein kinase 1, PKC; protein kinase C, PLC-β2; 1-phosphatidylinositol-4,5-bisphosphate phosphodiesterse β-2, PLK4; polo like kinase 4 or serine/threonine-protein kinase, PTEN; phosphatase and tensin homolog, PUMA; p53 upregulated modulator of apoptosis, Rab 1; Rab GTPases, RAF-B; v-Raf murine sarcoma viral oncogene homolog B, RANKL; receptor activator of nuclear factor kappa-B ligand, Rb-1; retinoblastoma-1, RUNX2; Runt-related transcription factor-2, SHH; sonic hedgehog, SMAD4; mothers against decapentaplegic, drosophila homolog 4, SOD-1; superoxide dismutase-1, SOS-1; son of sevenless homolog 1, SP-1; specificity protein 1, SRC1; steroid receptor coactivator-1, STAT3; signal transducer and activator of transcription-3, SVCT2; sodium-dependent vitamin C transporter 2, TERT; human telomerase reverse transcriptase, TGase-2; transglutaminase 2, TGF-β1; transforming growth factor-β1, TNFα; tumor necrosis factor-α, VCAM-1; vascular cell adhesion molecule-1, VE-cadherin; vascular endothelial cadherin, VEGF-A vascular endothelial growth factor A, VEGFR2; vascular endothelial growth factor receptor 2, p-VEGFR2; vascular endothelial growth factor receptor 2 (Y951), vWF; von Willebrand factor, Wnt1; proto-oncogene protein Wnt-1, YAP; Yes-associated protein.

20) at room temperature for 1 h. The supernatant of the antibody-incubated column was removed, and the immunoprecipitation procedures were performed.

Briefly, each protein sample was mixed with 5 mL of binding buffer (150mM NaCl, 10mM Tris pH 7.4, 1mM EDTA, 1mM EGTA, 0.2mM sodium vanadate, 0.2mM PMSF and 0.5% NP-40) and incubated in an antibody-bound protein A/G agarose bead column on a rotating stirrer at room temperature for 1 h. After multiple washing of the columns with phosphate-buffered saline solution in a graded NaCl concentration range (0.15–0.3M), the target proteins were eluted with 250μL of IgG elution buffer (Pierce, USA). The immunoprecipitated proteins were analyzed using a precision HPLC unit (1100 series, Agilent, Santa Clara, CA, USA) equipped with a reverse-phase column and a micro-analytical UV detector system (SG Highteco, Hanam, Korea). Column elution was performed using 0.15M NaCl/20% acetonitrile solution at 0.4 mL/min for 30 min, 30˚C, and the proteins were detected using a UV spectrometer at 280 nm. The control and experimental samples were run sequentially to allow comparisons. For IP-HPLC, the whole protein peak areas (mAU*s) were calculated mathematically using an analytical algorithm (see S1 Data) by subtracting the negative control antibody peak areas, and protein expression levels (concentrations, mAU) were compared and normalized using the square roots of protein peak areas. The ratios of the protein levels between the experimental and control groups were calculated and plotted. Protein expressional changes of less than ±5%, ±5–10%, ±10–20%, or over ±20% changes were described as minimal, slight, significant, or marked, respectively [19, 30–32].

When the IP-HPLC results were compared with the western blot data of the cytoplasmic housekeeping protein (β-actin), the former showed minute error ranges less than ±5% which was acceptable for statistical analysis. In contrast, the latter showed a large error range of more than 20%, making it almost impossible to analyze them statistically (see S2 Data). Therefore, the present study performed western blot using only selected antisera for cell differentiation (E-cadherin, VE-cadherin, and TGF-β1), apoptosis (c-caspase 3, PARP-1, and c-PARP-1), and endoplasmic reticulum (ER) stresses (eIF2AK3, eIF2α, ATF4, ATF6, GADD153, and LC3) to confirm their expression in 4HR-treated HUVECs. Otherwise, IP-HPLC was mostly utilized to analyze the global protein expression changes in cells statistically.

## Statistical analysis

Proportional data (%) of the experimental and control groups were plotted on line graphs and star plots. Their analyses were repeated two to six times until the standard deviations reached ≤±5%. The line graphs revealed time-dependent expression changes between the relevant proteins, and the star plots revealed the different expression levels of all proteins examined. The results were analyzed by measuring the standard error (s $= \pm\sqrt{\frac{a^2}{n}}$). The expression of the control housekeeping proteins, i.e., $\beta$-actin, $\alpha$-tubulin, and glyceraldehyde 3-phosphate dehydrogenase (GAPDH) was non-responsive (≤5%) to 8, 16, or 24 h of 4HR treatment.

## Results

### Proliferation index by direct cell counting assay

4HR-treated HUVECs, which were cultured on two-well culture slide dishes, showed decreases in cell number depending on time, at 8, 16, and 24 h (Fig 1). The number of HUVECs observed at x200 magnification was 65.9±5.46 at 0 h, as a control, 63.2±5.25 at 8 h, 56.4±6.47 at 16 h, and 51.7±6.38 at 24 h after the 4HR treatment (Fig 1A–1D and 1I). In particular, the 4HR-treated HUVECs contained many small vacuoles in their cytoplasm compared

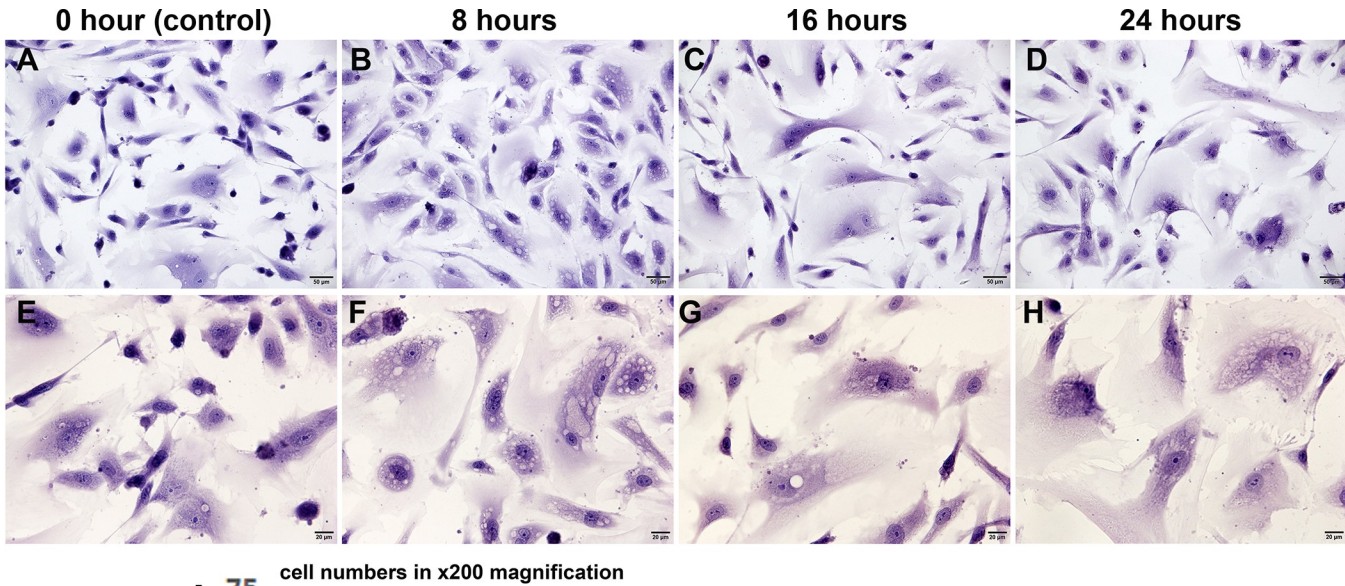

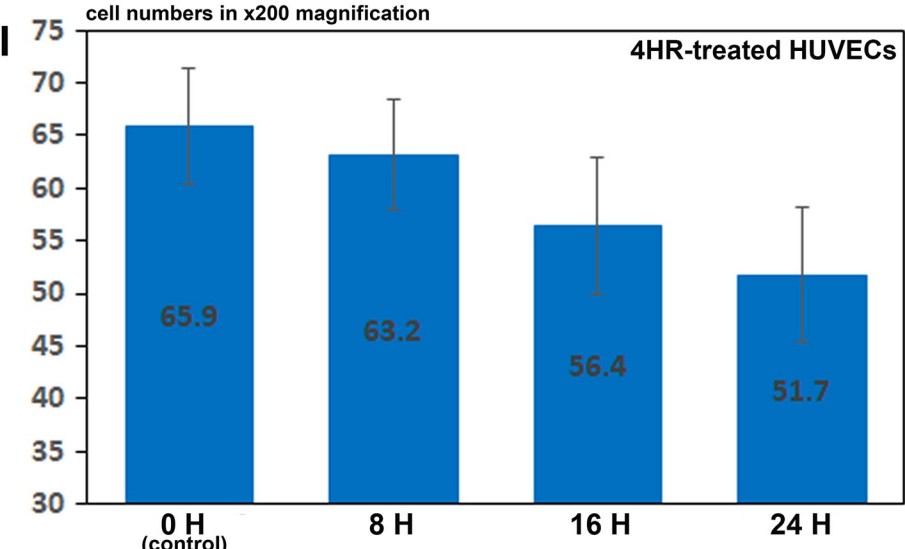

**Fig 1. Proliferation index of HUVECs by direct cell counting assay.** The 4HR-treated HUVECs showed decreases in cell number depending on time, at 0, 8, 16, and 24 hours (A-D and I), and contained many small vacuoles in their cytoplasm (E-H). Panels A, B, C, and D, are at x200 magnification; panel E, F, G, and H are at x400 magnification.

to the untreated controls, and these small vacuoles were similar to autophages in the histology observation (Fig 1E–1H).

### Immunocytochemical observation

Regarding the protein expression relevant to endothelial cell differentiation, the immunostainings of E-cadherin and VE-cadherin, cell adhesion molecules forming cadherin-catenin complex, were conspicuously positive in the 4HR-treated HUVECs compared to the untreated control cells. In particular, both E-cadherin and VE-cadherin were localized at the nuclei of 4HR-treated HUVECs at 16 and 24 h (Fig 2A and 2B). The immunoreaction of TGF-β1, a multifunctional protein exerting a role in cell growth, proliferation, differentiation, and apoptosis, increased in 4HR-treated cells at 8, 16, and 24 h compared to the untreated control cells. TGF-β1 was usually positive in the cytoplasm of cells (Fig 2C).

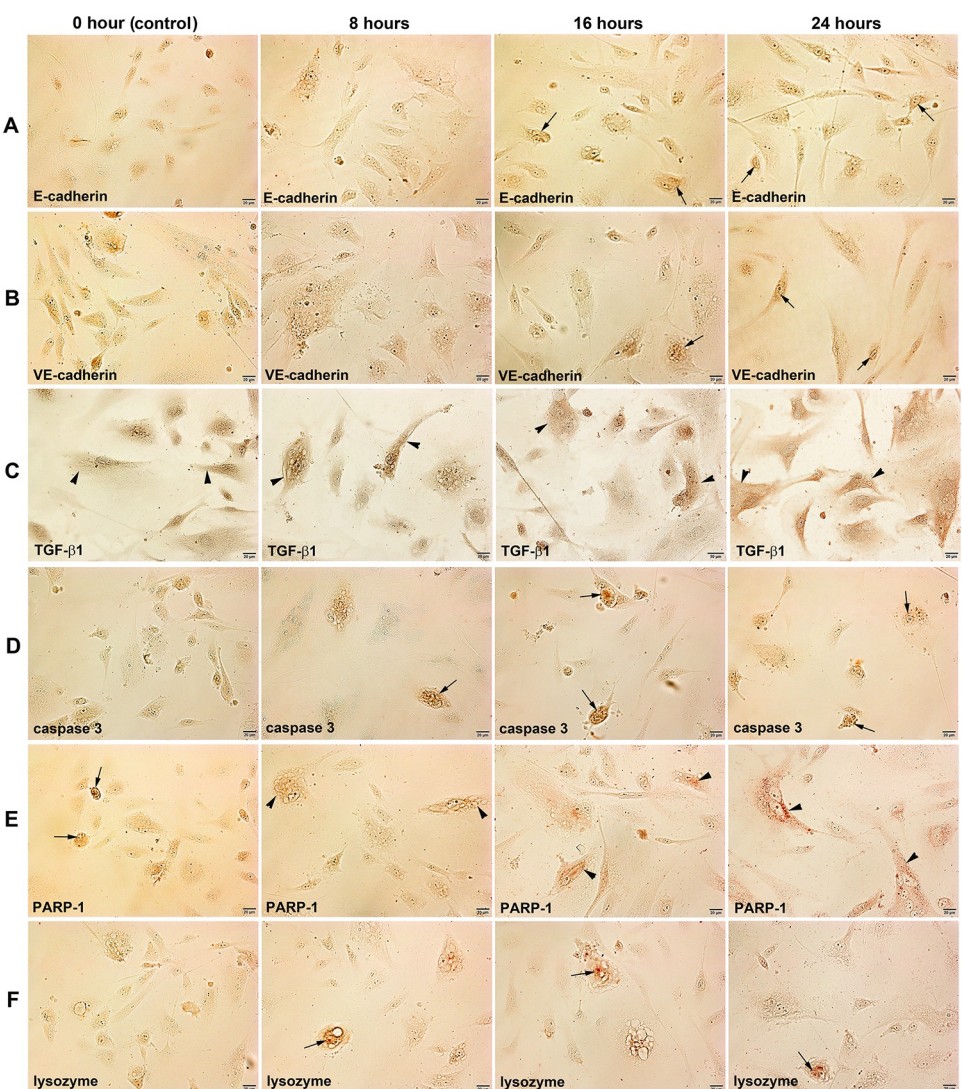

**Fig 2.** Immunocytochemical staining of E-cadherin (A), VE-cadherin (B), TGF-β1 (C), caspase 3 (D), PARP-1 (E), and lysozyme (F) in HUVECs after 4HR treatment for 0, 8, 16, 24 hours. Noted the cytoplasmic localization (arrow heads) and nuclear localization (arrows) of different immunoreactions.

Caspase 3, an apoptosis executing enzyme, was strongly positive in 4HR-treated cells compared to the untreated control cells, and its immunoreaction was localized at the nuclei (Fig 2D). PARP-1, a highly error-prone DNA repair pathway microhomolgy-mediated end joining enzyme, was usually positive in the nuclei of untreated control cells, but its immunoreaction was increased gradually in the cytoplasm of 4HR-treated cells at 8, 16, and 24 h (Fig 2E). Lysozyme, a cationic muiramidase, was diffusely positive in the cytoplasm of untreated control cells, but its immunoreaction was localized at the nuclei of 4HR-treated cells at 8, 16, and 24 h (Fig 2F).

The immunoreaction of eIF2AK3, a protein kinase R (PKR)-like endoplasmic reticulum kinase (PERK) that can inactivate eIF2α, increased gradually in 4HR-treated cells at 8, 16, and 24 h compared to the untreated control cells. eIF2AK3 was diffusely localized at the cytoplasm and nuclei of cells (Fig 3A). eIF2α, a regulator of global translation in response to cellular stress, was weakly positive in the untreated control cells, but it increased gradually in 4HR-

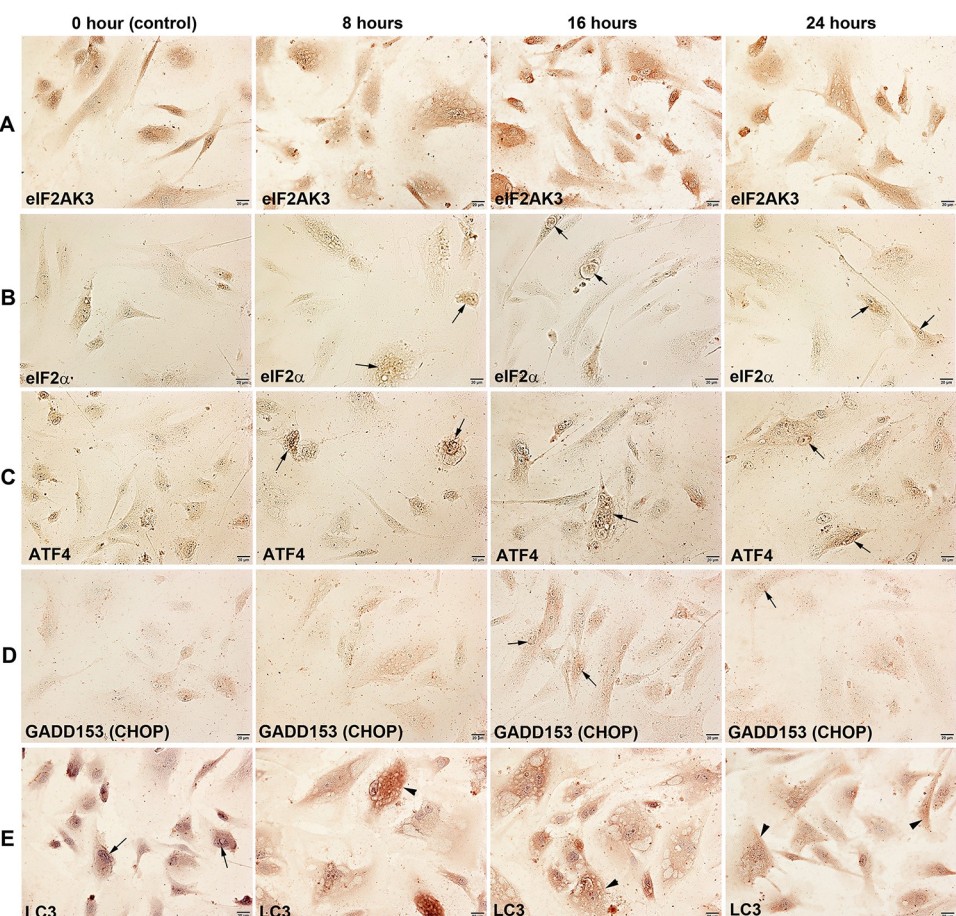

**Fig 3.** Immunocytochemical staining of eIF2AK3 (PERK) (A), eIF2α (B), ATF4 (C), GADD153 (CHOP) (D), and LC3 (E) in HUVECs after 4HR treatment for 0, 8, 16, 24 hours. Noted the cytoplasmic localization (arrow heads) and nuclear localization (arrows) of different immunoreactions.

treated cells at 8, 16, and 24 h (Fig 3B). ATF4, a master transcription factor during the integrated stress response, was weakly positive in the untreated control cells but became strongly positive and localized at the nuclei of 4HR-treated cells at 8, 16, and 24 h (Fig 3C).

GADD153 (CHOP or DDIT3), a DNA damage-inducible transcript 3 pro-apoptotic transcription factor, was weakly positive in the untreated control cells, but its immunoreaction increased slightly in the nuclei of 4HR-treated cells at 8, 16, and 24 h (Fig 3D). LC3, a biomarker of autophagosomes, was strongly positive in the nuclei but weak in the cytoplasm of the untreated cells. The immunoreaction of LC3 was observed in both the cytoplasm and nuclei of HUVECs, and became higher in 4HR-treated cells at 8, 16, and 24 h, and localized at the cytoplasm of cells but sparse in the nuclei (Fig 3E).

## Western blot detection for selected proteins

Protein expression in 4HR-treated HUVECs was confirmed by examining some selected proteins by western blot analysis. Proteins relevant to endothelial cell differentiation, both E-cadherin and VE-cadherin were upregulated slightly at 8 and 16 h but downregulated at 24 h compared to the untreated controls, and TGF-β1, a multifunctional protein relevant to cellular differentiation and apoptosis, was consistently upregulated and showed intense expression at 24 h (Fig 4A).

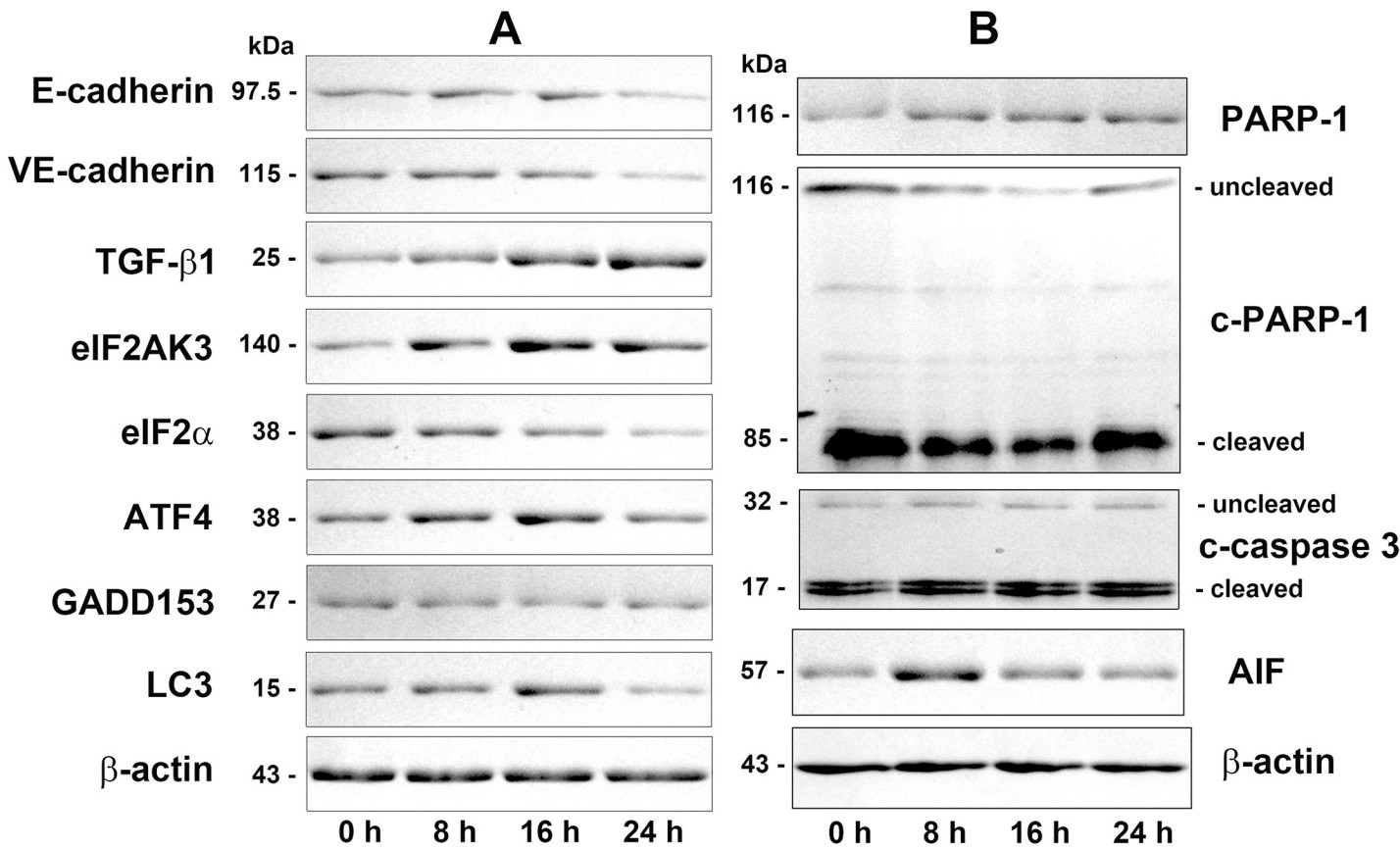

**Fig 4. Western blot analysis for the protein expression for endothelial cell differentiation (E-cadherin, VE-cadherin, and TGF-β1), ER stresses (eIF2AK3 (PERK), eIF2α, ATF4, GADD153 (CHOP), and LC3), and apoptosis (c-caspase 3, PARP-1, c-PARP-1, and AIF) in HUVECs at 0, 8, 16, and 24 h after 4HR treatment.** c-caspase 3 polyclonal antibody (PoAb) against amino-terminal residues adjacent to Asp 175 in human caspase 3 detected strong cleaved caspase 3 bands (17 kDa) but weak uncleaved caspase 3 bands (32 kDa), PARP-1 PoAb against C-terminal amino acids (764–1014 aa) of human PARP-1 detected only full length PARP-1 bands (116 kDa), and c-PARP-1 PoAb against a short amino acid sequence containing Gly 215 of human PARP-1 detected strong cleaved PARP-1 bands (85 kDa) but weak uncleaved PARP-1 bands (116 kDa).

eIF2AK3 (PERK), eIF2α, ATF4, and GADD153, contributing eIF2AK3/eIF2α/ATF4/ GADD153 signaling for ER stresses, increased or decreased variably, and the expression of eIF2AK3 and ATF4 increased significantly at 8, 16, and 24 h after 4HR treatment. GADD153 expression changed minimally after the 4HR treatment, while eIF2α expression decreased slightly. On the other hand, the expression of LC3, which plays a central role in the autophagy pathway, was significantly higher at 8 and 16 h after the 4HR treatment (Fig 4A).

Regarding 4HR-induced apoptosis of HUVECs, c-caspase 3 was increased gradually at 8, 16, and 24 h compared to the untreated control. PARP-1 also gradually increased at 8, 16, and 24 h, while c-PARP-1 decreased at 8 h but increased in 24 h. In particular, AIF was markedly increased at 8 h, and increased slightly at 16 and 24 h compared to the untreated control (Fig 4B).

## Effects of 4HR on the expression of proliferation-related proteins

HUVECs treated with 4HR for 8, 16, or 24 h exhibited gradual decreases in the levels of proliferation-activating proteins, Ki-67 by 15.2% at 48 h, proliferation cell nuclear antigen (PCNA) by 11.6% at 16 h, mitotic protein monoclonal 2 (MPM2) by 20.6% at 24 h, cyclin-dependent

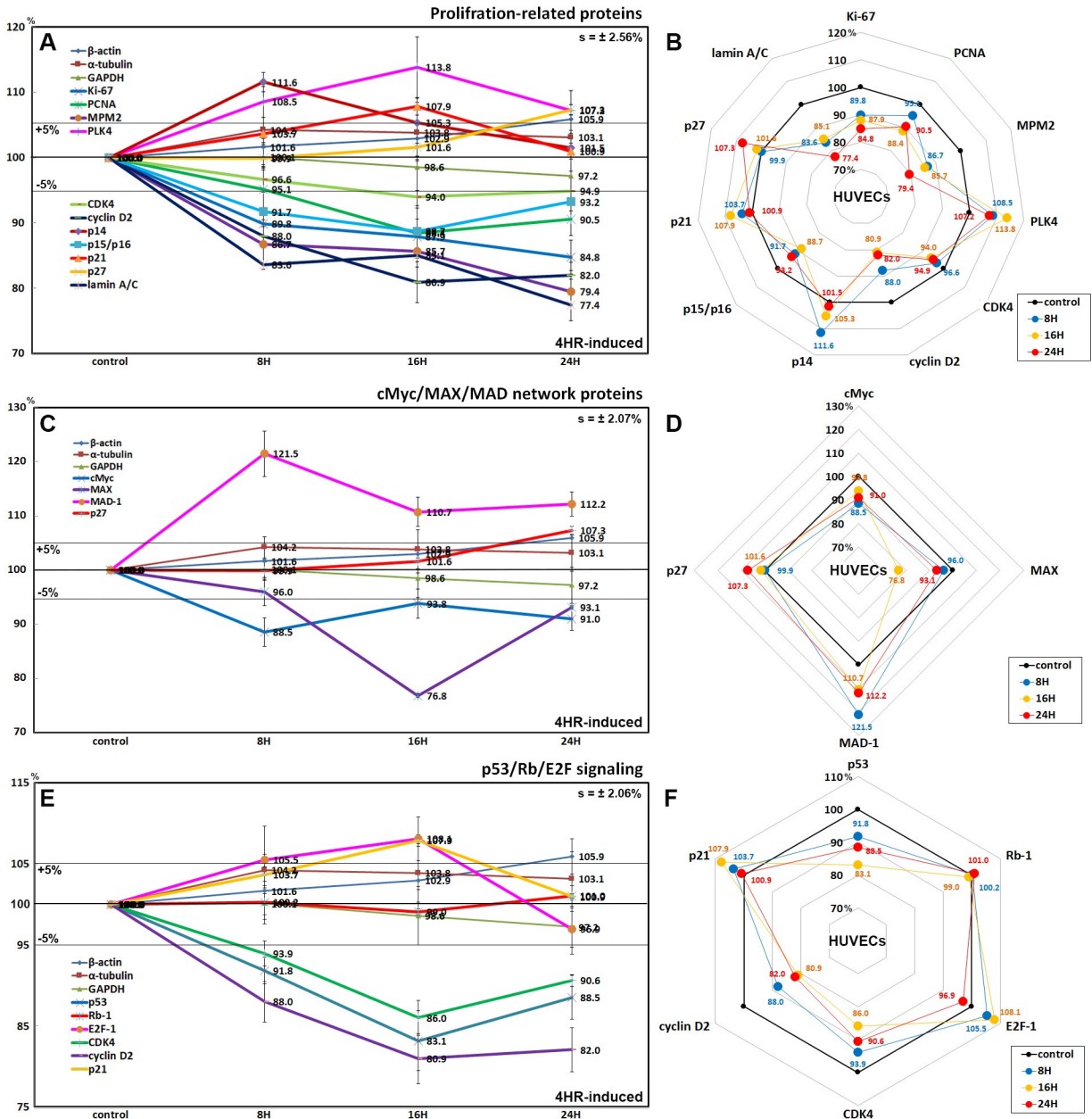

**Fig 5.** Expression of proliferation-related proteins (A and B), cMyc/MAX/MAD network proteins (C and D), and p53/Rb/E2F signaling proteins (E and F) in 4HR-treated HUVECs as determined by IP-HPLC. Line graphs, A, C, and E show protein expression on the same scale (%) versus culture time (8, 16, or 24 h), whereas the star plots (B, D, and F) showed the differential expression levels of the proteins at 8, 16, or 24 h after 4HR administration on the appropriate scales (%). The thick black line, untreated controls (100%); the orange, pink, and red dots show differential protein levels after 4HR administration for 8, 16, or 24 h, respectively.

kinase 4 (CDK4) by 6% at 16 h, cyclin D2 by 19.1% at 16 h, and lamin A/C by 22.6% at 24 h. In contrast, the levels of proliferation-inhibiting proteins such as p14, p21, and p27 were increased by 11.6% at 8 h, 7.9% at 16 h, and 7.3% at 24 h, respectively, compared to the untreated controls. On the other hand, the expression of polo-like kinase 4 (PLK4) increased by 13.8% at 16 h, while p15/16 expression decreased by 11.3% at 16 h (Fig 5A and 5B).

### Effects of 4HR on the expression of cMyc/MAX/MAD network proteins

4HR decreased the expression of cMyc and MAX, forming Myc-MAX heterodimer complex, by 11.5% and 23.2% at 8 h and 16 h, respectively, after 4HR administration compared to the untreated controls. In contrast, MAD-1 expression was increased by 21.5% at 8 h, and this elevated level was maintained at 12.2% after 24 h. Concomitantly, the expression of p27, a rate-limiting cell cycle target of cMyc, increased gradually by 7.3% at 24 h (Fig 5C and 5D).

### Effects of 4HR on the expression of p53/Rb/E2F signaling proteins

4HR decreased the expression of p53 in HUVECs by 16.9% and 11.5% at 16 h and 24 h, respectively, compared to the untreated controls, while p21 (cyclin-dependent kinase inhibitor 1) expression was increased by 7.9% at 16 h. Although Rb-1 expression was almost unaffected ($\leq$1%), the expression of E2F-1 (an objective transcription factor) increased by 8.1% after 16 h but decreased by 3.1% at 24 h after administration. In contrast, the levels of CDK4 and cyclin D2 expression decreased by 14% and 19.1% at 16 h, respectively (Fig 5E and 5F).

### Effects of 4HR on the expression of Wnt/β-catenin signaling proteins

4HR reduced the protein expression of Wnt1 (by 9.6% at 24 h), β-catenin (by 6.9% at 24 h), adenomatous polyposis coli (APC; by 20.6% at 8 h), and snail (a transcription factor repressing the expression of the adhesion molecules, by 26.8% at 8 h) compared to the untreated controls. The expression of T-cell factor 1 (TCF-1, a transcription factor) was also reduced by 6.7% at 16 h. In contrast, the expression of E-cadherin (a type I transmembrane protein stabilized by β-catenin) increased slightly by 6.2% at 16 h, and the expression of VE-cadherin (vascular endothelial cadherin) increased markedly by 23.6% at 16 h (Fig 6A and 6B).

### Effects of 4HR on the expression of epigenetic modification-related proteins

4HR significantly increased the expression of histone H1 (by 29.7% at 8 h), histone deacetylase 10 (HDAC-10, by 32.3% at 8 h), DNA (cytosine-5)-methyl transferase 1 (DNMT1, by 23.7% at 8 h), and DNA methyl transferase 1-associated protein 1 (DMAP1, by 31.7% at 24 h), and slightly increased the expression of p300/CBP-associated factor K (lysine) acetyl-transferase 2B (PCAF, by 5.2% at 24 h). On the other hand, lysine-specific demethylase 4D (KDM4D) was decreased by 11.3% at 8 h but increased by 15.3% at 24 h, and methyl-CpG binding domain 4 (MBD4) was consistently decreased by 26.3% at 16 h and 27.8% at 24 h (Fig 6C and 6D).

### Effects of 4HR on the expression of translation-related proteins

The protein translation-related protein levels decreased gradually in HUVECs treated with 4HR. Deoxyhypusine hydroxylase (DOHH) expression was decreased by 19.8% and 11.8% at 16 and 24 h, respectively. Deoxyhypusine synthase (DHS) expression was decreased by 22.1% at 24 h. The protein expression of objective factors of protein translation was reduced by 4HR; eukaryotic translation initiation factor 5A-1 (eIF5A-1) by 5.4% at 24 h. In contrast, the protein expression related to ER stresses were increased by 4HR: eukaryotic translation initiation factor 2-α kinase 3 (eIF2AK3, PERK) and p-eIF2AK3 by 18.4% at 16 h and 28.1% at 24 h, respectively; eIF2α was downregulated by 7.8% at 24 h, while p-eIF2AK3 was upregulated by 6.6% at 16 h. (Fig 6E and 6F).

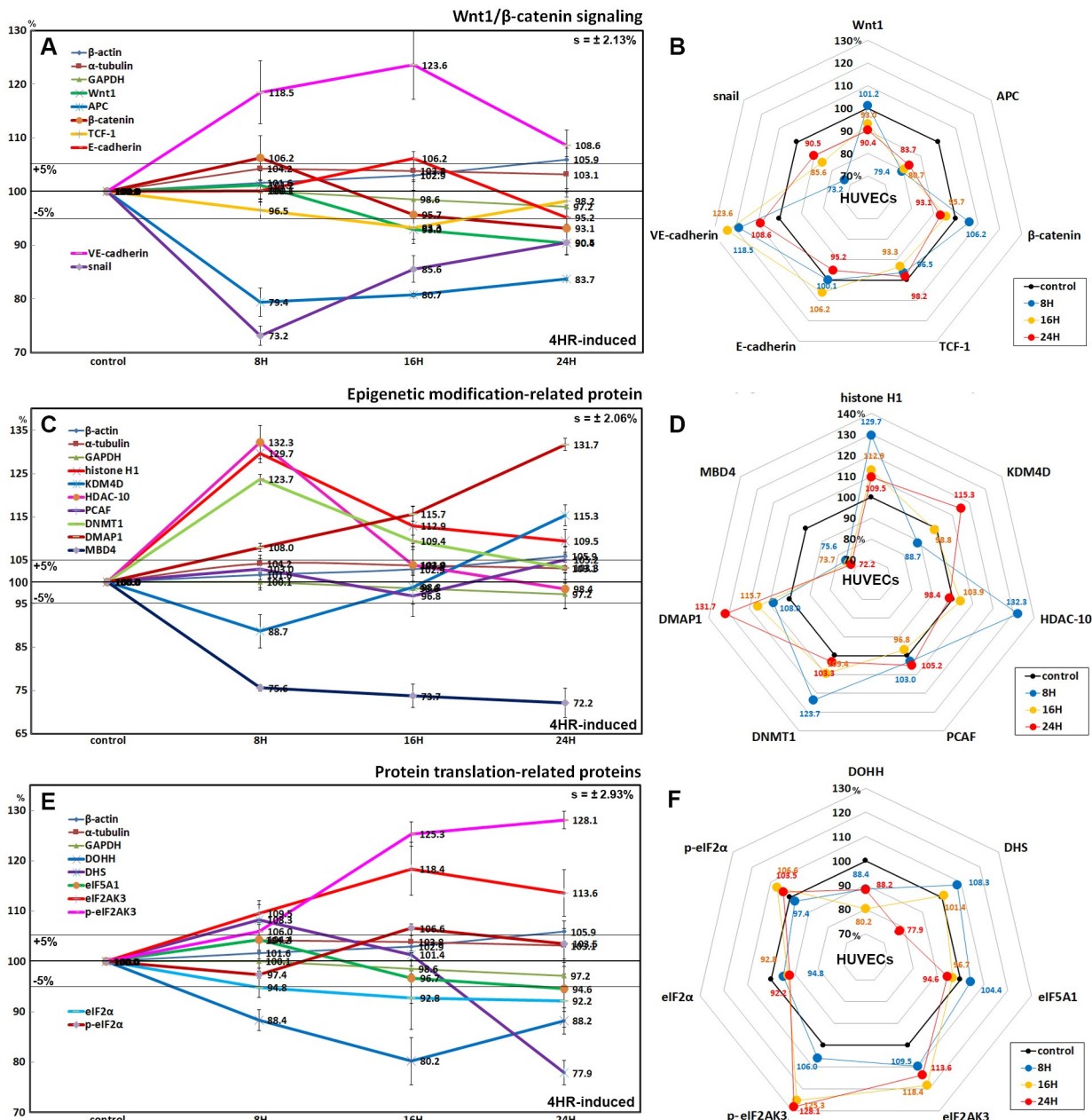

**Fig 6.** Expression of Wnt/β-catenin signaling proteins (A and B), epigenetic modification-related proteins (C and D), and protein translation-related proteins (E and F) in 4HR-treated HUVECs as determined by IP-HPLC. Line graphs, A, C, and E show protein expression on the same scale (%) versus culture time (8, 16, or 24 h), whereas the star plots (B, D, and F) showed the differential expression levels of the proteins at 8, 16, or 24 h after 4HR administration on the appropriate scales (%). The thick black line, untreated controls (100%); the orange, pink, and red dots show differential protein levels after 8, 16, or 24 h of 4HR administration, respectively.

## Effects of 4HR on the expression of growth factor-related proteins

HUVECs after 4HR administration showed marked increases in the expression of transforming growth factor-β1 (TGF-β1, by 29.3% at 24 h), TGF-β2 (7.3% at 8 h), TGF-β3 (22.3% at 24 h), SMAD2/3 (27.1% at 16 h), SMAD4 (13.4% at 8 h), p-SMAD4 (13.3% at 16 h), HGFα (25.8% at 16 h), Met (19.8% at 16 h), IGF-1 (17.2% at 16 h), HER1 (20.2% at 8 h), and slight

increases in the expression of fibroblast growth factor-7 (FGF-7, 7.4% at 16 h) and connective tissue growth factor (CTGF, 8.4% at 24 h) compared to the untreated controls, but decreases in the expression of FGF-1 (16.4% at 16 h), FGF-2 (6.1% at 24 h), growth hormone (GH, 4.2% at 8 h), growth hormone releasing hormone (GHRH, 9.7% at 24 h), insulin-like growth factor 2 receptor (IGFIIR, 17.5% at 16 h), platelet-derived growth factor-A (PDGF-A, 5.1% at 16 h), HER2 (17% at 8 h), and estrogen receptor-β (ERβ, 20.5% at 16 h) (Fig 7A and 7B).

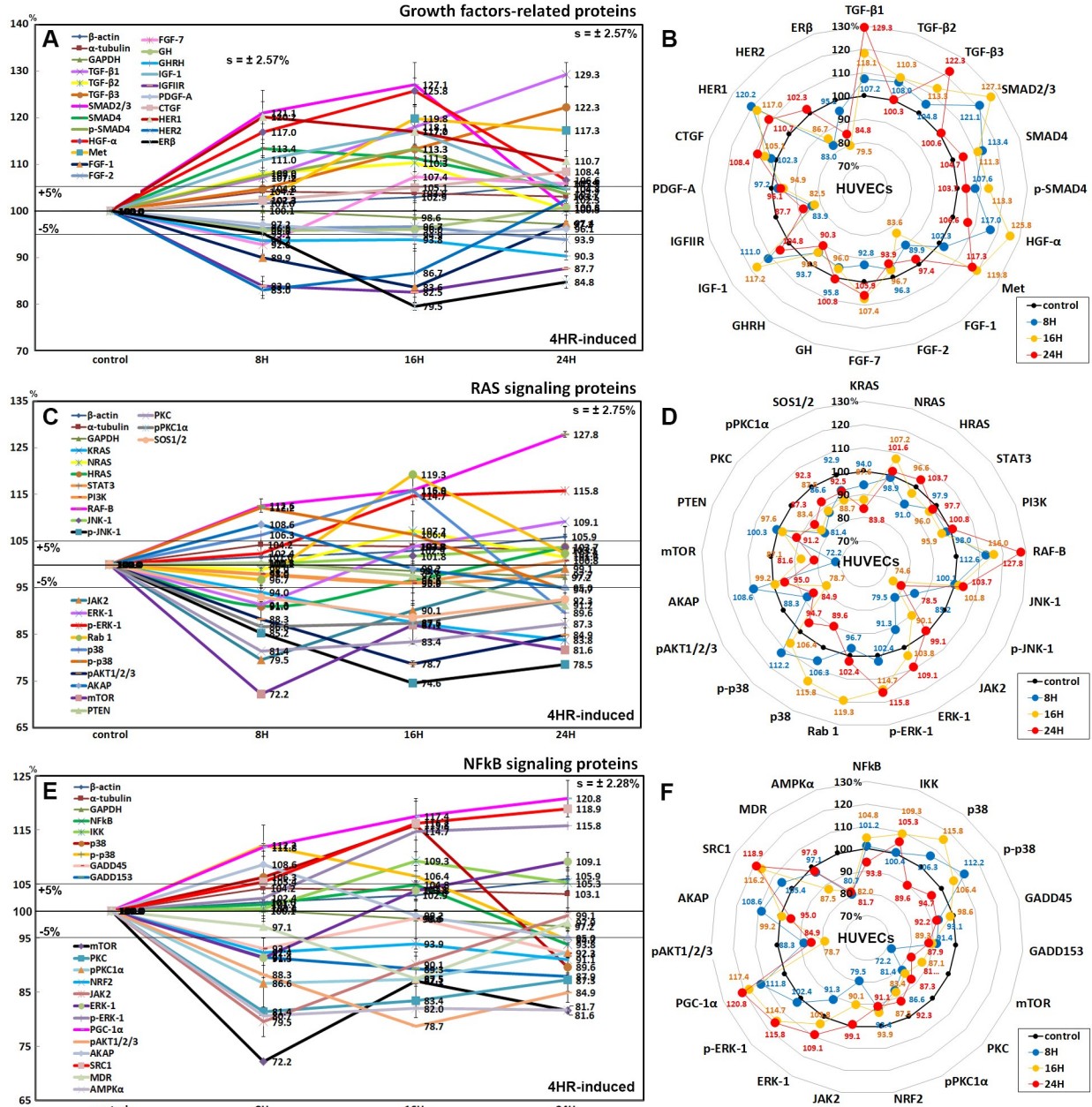

**Fig 7.** Expression of growth factors (A and B), RAS signaling proteins (C and D), and NFkB signaling proteins (E and F) in 4HR-treated HUVECs as determined by IP-HPLC. Line graphs, A, C, and E show protein expression on the same scale (%) versus culture time (8, 16, or 24 h), whereas the star plots (B, D, and F) showed the differential expression levels of the proteins at treatment at 8, 16, or 24 h after 4HR administration on the appropriate scales (%). The thick black line, untreated controls (100%); the orange, pink, and red dots show differential protein levels after 4HR administration for 8, 16, or 24 h, respectively.

## Effects of 4HR on the protein expression of the RAS signaling proteins

The effects of the treatment with 4HR for 24 h on the expression of the RAS signaling proteins in HUVECs were variable. KRAS expression decreased gradually by 16.2% at 24 h, HRAS expression decreased by 9% at 8 h but increased by 3.7% at 24 h compared to the untreated controls, whereas NRAS expression increased by 2% and 1.6% at 16 h and 24 h, respectively. Many upstream proteins were downregulated by 4HR administration: phosphorylated c-Jun N-terminal kinase-1 (p-JNK-1, by 25.4% at 16 h), which is responsible for the responses to stressors, such as cytokines, ultraviolet irradiation, heat shock, and osmotic shock, Janus kinase 2 (JAK2, a non-receptor tyrosine kinase implicated in signaling by members of the type II cytokine receptor family, 20.5% at 8 h), pAKT1/2/3 (the critical mediator of growth factor-induced signals; Thr 308, 21.3% at 16 h), A-kinase anchoring proteins (AKAP, 5% at 24 h), mammalian target of rapamycin (mTOR, 27.8% at 8 h), phosphatase and tensin homolog (PTEN, 8.8% at 24 h), protein kinase C (PKC, 18.6% at 8 h), pPKC1α (13.4% at 8 h), and a son of sevenless homolog 1/2 (SOS1/2, 11.3% at 16 h). Some downstream proteins were upregulated by 4HR: serine/threonine-protein kinase RAF-B (27.8% at 24 h), extracellular signal-regulated kinase 1 (ERK-1, 9.1% at 24 h), p-ERK-1 (15.8% at 24 h), GTPases Rab1 (19.3% at 16 h), p38 (15.8% at 16 h), and p-p38 (12.2% at 8 h). On the other hand, the expression of signal transducer and activator of transcription 3 (STAT3), phosphatidylinositol 3-kinase (PI3K), and c-Jun N-terminal kinases-1 (JNK-1) were affected minimally by 4HR (≤5%) (Fig 7C and 7D).

## Effects of 4HR on the expression of NFkB signaling proteins

4HR had different effects on the expression of nuclear factor kappa-light-chain-enhancer of activated B cells (NFkB) signaling proteins. The expression of NFkB was reduced slightly by 6.2% at 24 h compared to the untreated controls. In contrast, the expression of ikappaB kinase (IKK), p38, and p-p38, which are negative regulators of the NFkB function, were increased by 9.3%, 15.8%, and 12.2% at 16 h, 16 h, and 8 h, respectively.

4HR reduced the protein expression of downstream proteins of NFkB signaling; growth arrest and DNA damage 45 (GADD45, by 7.8% at 24 h), GADD153 (12.1% at 24 h), mTOR (by 27.8% at 8 h), PKC (18.6% at 8 h), pPKC1α (13.4% at 8 h), nuclear factor (erythroid-derived 2)-like 2 (NRF2, by 8.9% at 24 h), JAK2 (20.5% at 8 h), pAKT1/2/3 (21.3% at 16 h), AKAP (by 5% at 24 h), multiple drug resistance (MDR, 12.5% at 16 h), and 5' AMP-activated protein kinase α (AMPKα, by 15.9% at 8 h). In contrast, it increased the expression of ERK-1 (9.1% at 24 h), pERK-1 (15.8% at 24 h), peroxisome proliferator-activated receptor-gamma coactivator 1-α (PGC-1α, by 20.8% at 24 h), and steroid receptor coactivator-1 (SRC1, by 18.9% at 24 h) (Fig 7E and 7F).

## Upregulated inflammatory proteins by the 4HR treatment

Proteins that regulate inflammatory reactions were generally upregulated by 4HR compared to the untreated controls: IL-1 (by 15.3% at 24 h), IL-10 (15.1% at 16 h), IL-12 (8.7% at 16 h), cathepsin K (13% at 8 h), lysozyme (25.5% at 24 h), granzyme B (33.9% at 24 h), lactoferrin (37.9% at 24 h), macrophage colony-stimulating factor (M-CSF, 17.6% at 8 h), β-defensin-1 (19.8% at 24 h), β-defensin-2 (25.5% at 16 h), β-defensin-3 (14.9% at 8 h), CD28 (8.2% at 16 h), programmed cell death protein-1/1 (Pdcd-1/1, 12% at 8 h), PECAM-1 (19.8% at 8 h), homing cell adhesion molecule (HCAM, 5.8% at 8 h), ICAM-1 (intercellular adhesion molecule 1, 17.7% at 16 h), versican (9.3% at 8 h), cyclooxygenase 1 (COX-1, 36.2% at 16 h), COX-2 (27% at 16 h), and kininogen-1 (16.6% at 8 h) (Fig 8A and 8B).

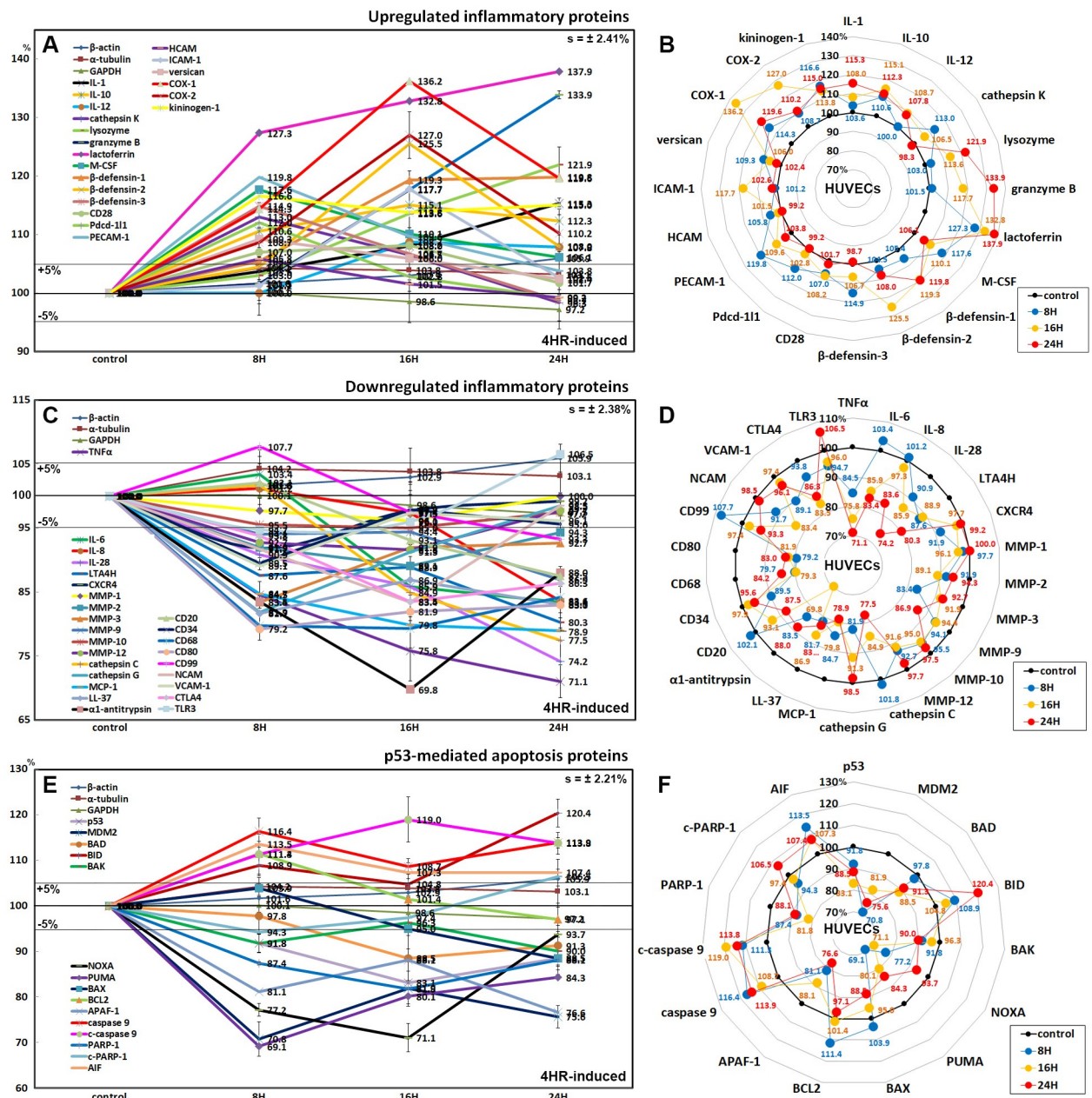

**Fig 8.** Expression of upregulated inflammatory proteins (A and B), downregulated inflammatory proteins (C and D), and p53-mediated apoptosis-related proteins (E and F) in 4HR-treated HUVECs as determined by IP-HPLC. Line graphs, A, C, and E show protein expression on the same scale (%) versus culture time (8, 16, or 24 h), whereas star plots (B, D, and F) showed the differential expression of the proteins at 8, 16, or 24 h after 4HR administration on the appropriate scales (%). The thick black line; untreated controls (100%); the orange, pink, and red dots show differential protein levels after 4HR administration for 8, 16, or 24 h, respectively.

## Downregulated inflammatory proteins by the 4HR treatment

Inflammatory proteins associated with the immediate inflammatory reactions were downregulated by 4HR: tumor necrosis factor α (TNFα, 28.9% at 24 h), IL-6 (16.6% at 24 h), IL-28 (25.8% at 24 h), leukotriene A4 hydrolase (LTA4H, 19.7% at 24 h), C-X-C chemokine receptor type 4 (CXCR4, 8.1% at 8 h), MMP-1 (3.9% at 16 h), MMP-2 (10.9% at 16 h), MMP-3 (16.6% at 8 h), MMP-9 (13.1% at 24 h), MMP-10 (5% at 16 h), MMP-12 (8.4% at 16 h), cathepsin C

(22.5% at 24 h), cathepsin G (18.1% at 8 h), monocyte chemotactic protein-1 (MCP-1, 21.1% at 24 h), LL-37 (18.3% at 8 h), α1-antitrypsin (30.2% at 16 h), CD20 (12.5% at 24 h), CD34 (10.5% at 8 h), CD68 (20.7% at 16 h), CD80 (20.8% at 8 h), CD99 (6.7% at 24 h), NCAM (16.6% at 16 h), VCAM-1 (10.9% at 8 h), cytotoxic T lymphocyte-associated protein-4 (CTLA4, 6.5% at 16 h), and toll-like receptor 3 (TLR3, 6.3% at 8 h) (Fig 8C and 8D).

## Effects of 4HR on the expression of p53-mediated apoptosis-related proteins

4HR affected the expression of p53-mediated apoptosis-related proteins, particularly the expression of the p53 protein, which was decreased by 16.9% at 16 h after treatment. 4HR also downregulated the expression of pro-apoptotic proteins: BCL2-associated death promoter (BAD, by 11.5% at 16 h), BCL2 homologous antagonist/killer (BAK, 10% at 24 h), NOXA (28.9% at 16 h), PUMA (30.9% at 8 h), apoptosis regulator BAX (11.5% at 24 h), and apoptotic protease activating factor 1 (APAF-1, 23.4% at 24 h). In contrast, 4HR upregulated the expression of B cell lymphoma 2 (BCL2, 11.4% at 8 h) and apoptosis-inducing factor (AIF, 13.5% at 8 h). On the other hand, the expression of the apoptosis executor proteins, caspase 9 and c-caspase 9, were upregulated by 13.9% and 19% at 24 h and 16 h, respectively. In contrast, the expression of PARP-1 (poly [ADP-ribose] polymerase 1) and c-PARP-1 were downregulated by 18.2% and 5.7% at 16 h and 8 h, respectively, but c-PARP-1 expression was upregulated by 6.5% at 24 h (Fig 8E and 8F).

## Effects of 4HR on the expression of FAS-mediated apoptosis-related proteins

HUVECs showed decreases in the expression of FAS-associated signaling proteins after 4HR administration. After treatment with 4HR for 24 h, the expression of FAS was decreased by 10.9% at 16 h. The expression of the FAS-associated protein with the death domain (FADD) was also decreased by 11.9% at 16 h, even though FASL expression was increased by 28.7% at 24 h. The expression of the FLICE-like inhibitory protein (FLIP) was increased by 29.7% at 24 h. In addition, 4HR-treated HUVECs showed elevated levels of apoptosis executor proteins: c-caspase 8 (by 8.6% at 8 h), c-caspase 10 (18.9% at 8 h), caspase 3 (by 9.8% at 16 h), c-caspase 3 (by 30.2% at 16 h), and BH3 interacting-domain death agonist (BID, by 20.4% at 24 h) (Fig 9A and 9B).

## Effects of 4HR on the expression of cell protection- and survival-related proteins

4HR-treated HUVECS showed reductions in the expression of cellular stress protection- and antioxidant-related proteins compared to the untreated controls, as follows: PKC (by 18.6% at 8 h), pPKC1α (13.4% at 8 h), pAKT1/2/3 (21.3% at 16 h), heat shock protein-70 (HSP-70, 12.6% at 8 h), HSP-90 (10.6% at 16 h), AMPKα (15.9% at 8 h), nitric oxide synthases-1 (NOS1, 11% at 8 h), heme oxygenase-1 (HO-1, 5.2% at 16 h), superoxide dismutase-1 (SOD-1, 8% at 16 h), sodium-dependent vitamin C transporter 2 (SVCT2, 16.1% at 8 h), and a transcription factor regulating the expression of antioxidant proteins (NRF2, 8.9% at 24 h). In contrast, there were elevations of the following: HSP-27 (up to 111.5% at 24 h), SOD-1 (13.6% at 8 h), and glutathione S-transferase ω 1 (GSTO1, by 117% at 8 h). Whereas 4HR consistently increased the expression of cellular maintenance proteins; LC3 (15.1% at 16 h), peroxisome proliferator-activated receptor gamma coactivator 1-alpha (PGC-1α, 20.8% at 24 h), telomerase reverse transcriptase (TERT, 28% at 24 h), SP-1 (17.8% at 24 h), SP-3 (22.4% at 8 h), leptin (16.4% at 24 h), and phosphoinositide-specific phospholipase C (PLC-β2, 15% at 16 h). On the

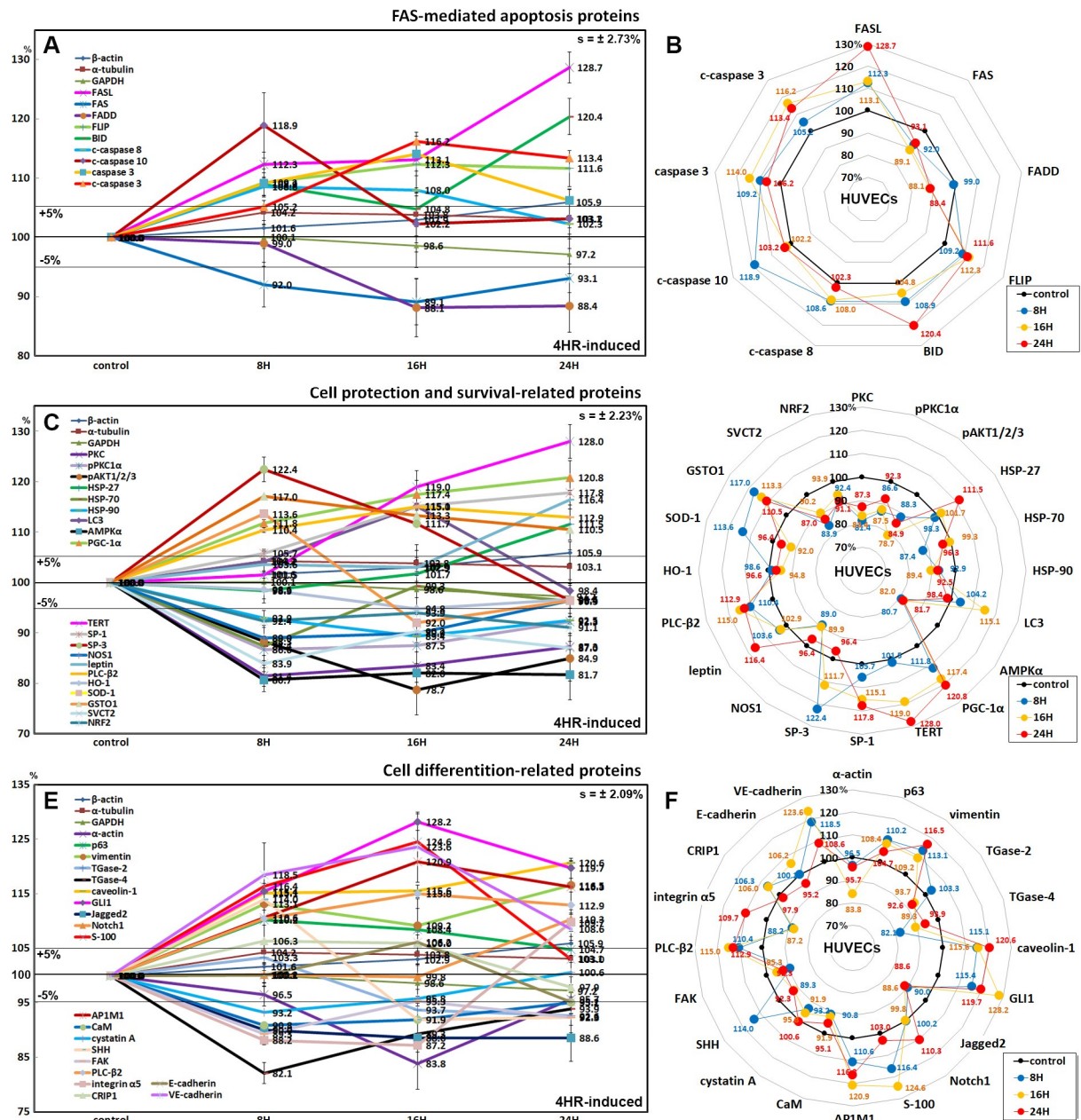

**Fig 9.** Expression of FAS-mediated apoptosis-related proteins (A and B), protection-related proteins (C and D), and differentiation-related proteins (E and F) in 4HR-treated HUVECs as determined by IP-HPLC. Line graphs, A, C, and E show protein expression patterns on the same scale (%) versus culture time (8, 16, or 24 h), whereas the star plots (B, D, and F) showed the differential expression levels of the proteins at 8, 16, or 24 h after 4HR administration on the appropriate scales (%). The thick black line, untreated controls (100%), the orange, pink, and red dots show differential protein levels after 4HR administration for 8, 16, or 24 h, respectively.

other hand, 4HR upregulated HSP-27 (up to 11.5% at 24 h) and glutathione S-transferase ω 1 (GSTO1, by 17% at 8 h) (Fig 9C and 9D).

## Effects of 4HR on the expression of differentiation-related proteins

4HR was found to influence the expression of cellular differentiation-related proteins positively or negatively in HUVECs. Some cytodifferentiation proteins were upregulated by 4HR,

i.e., p63 (by 10.2% at 8 h), E-cadherin (6.2% at 8 h), VE-cadherin (23.6% at 16 h), vimentin (16.5% at 24 h), caveolin-1 (20.6% at 24 h), GLI1 (28.2% at 16 h), Notch1 (10.3% at 24 h), S-100 (24.6% at 16 h), AP-1 complex subunit mu-1 (AP1M1, 20.9% at 16 h), sonic hedgehog (SHH, 14% at 8 h), PLC-β2 (15% at 16 h), integrin α5 (9.7% at 24 h), and cysteine-rich protein-1 (CyRP-1, 6.3% at 8 h). On the other hand, other cytodifferentiation proteins were downregulated by 4HR, i.e., α-actin (16.2% at 16 h), TGase-2 (7.4% at 24 h), TGase-4 (17.9% at 8 h), Jagged2 (11.4% at 16 h), calmodulin (CaM, 9.2% at 8 h), cystatin A (6.8% at 8 h), SHH (8.1% at 16 h), focal adhesion kinase (FAK, 10.7% at 8 h), and integrin α5 (11.8% at 8 h) (Fig 9E and 9F).

## Effects of 4HR on the expression of endoplasmic reticulum stress-related proteins in HUVECs

4HR-treated HUVECs showed an increase in protein expression related to ER stresses. Proteins contributing to ER stress signaling were upregulated by 4HR; eIF2AK3 and p-eIF2AK3, which function as an ER kinase (PERK), were increased by 18.4% and 28.1% at 16 h and 24 h, respectively, compared to the untreated controls, eIF2α and p-eIF2α, which are essential factors for protein synthesis also responsible for ER stresses, were decreased by 7.8% at 24 h and increased by 6.6% at 16 h, respectively, ATF4 and ATF6 (activating transcription factor 4 and 6), were increased by 30.2% at 24 h and 31.8% at 16 h, respectively. Subsequently, GADD153, which is a DNA damage-inducible pro-apoptotic transcription factor, was decreased by 12.1% at 24 h. The expression of LC3, an autophage microtubule-associated protein contributing to autophagosome biogenesis, was increased by 15.1% at 16 h. On the other hands, although HSP-70 chaperone, engaging in protein refolding, was decreased by 12.6% at 8 h, different proteins related to ER stresses including HSP-27 (preventing cell death induced by ER stresses), AIF (responding to ER stresses), AP1M1 (a trans-Golgi network clathrin-associated protein complex AP-1), endothelin-1 (inducing $Ca^{++}$ release from the ER), and PGC-1α (the master regulator of mitochondrial biogenesis, a key transcription factor involved in mediating the unfolded protein response) were increased by 11.5% at 24 h, 13.5% at 8 h, 20.9% at 16 h, 17.1% at 24 h, and 20.8% at 24 h, respectively (Fig 10A and 10B).

## Effects of 4HR on the expression of oncogenesis-related proteins in HUVECs

4HR-treated HUVECs showed less oncogenic potential than the untreated controls because 4HR downregulated the proteins reactive to oncogenic stress compared to the untreated controls as follows: PTEN (by 8.8% at 24 h), breast cancer type 1 susceptibility protein (BRCA1, 16.1% at 16 h), BRCA2 (12.8% at 8 h), a DNA repair enzyme that removes mismatched U or T (MBD4, 27.8% at 24 h), and a phosphoserine binding protein that regulates Cdc25C by sequestering it in the cytoplasm (14-3-3, 8.1% at 24 h). In addition, 4HR downregulated the expression of oncogenesis-related proteins, i.e., a negative regulator of apoptosis (survivin, 14.8% at 8 h), an anti-adhesive glycoprotein that contributes to tumor development and metastasis (mucin 4, 5.7% at 24 h), and a potent oncogene that binds to 14-3-3 (YAP, 20.6% at 8 h). On the other hand, 4HR upregulated the expression of tumor suppressor proteins, i.e., GTPase-activating protein that negatively regulates RAS/MAPK pathway activity (NF-1, 12.4% at 16 h), a serine/threonine-protein kinase recruited and activated by DNA double-strand breaks (ATM, 11.4% at 24 h), a suppressor of smoothened release, which signals cell proliferation (PTCH-1, 9.9% at 16 h), deleted in malignant brain tumors 1 protein, a glycoprotein that interacts between tumor cells and the immune system (DMBT1, 13.7% at 24 h), and a protective glycosylated phosphoprotein that binds to pathogens (mucin 1, 12.7% at 16 h). The expression

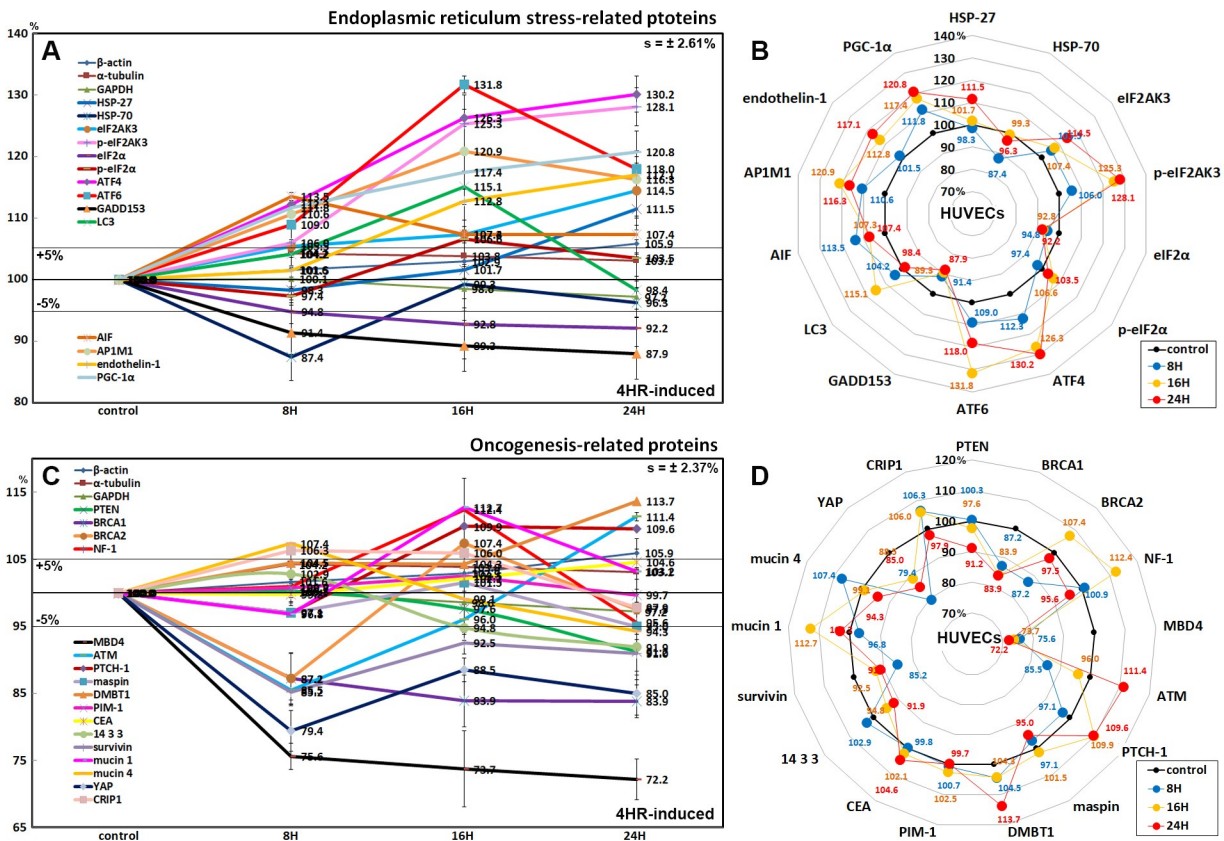

**Fig 10.** Expression of ER stress-related proteins (A and B) and oncogenesis-related proteins (C and D) in 4HR-treated HUVECs as determined by IP-HPLC. Line graphs, A and C show protein expression on the same scale (%) versus culture time (8, 16, or 24 h), whereas the star plots (B and D) showed the differential expression levels of the proteins at 8, 16, or 24 h after 4HR administration on the appropriate scales (%). Thick black line, untreated controls (100%); the orange, pink, and red dots show differential protein levels after 4HR administration for 8, 16, or 24 h, respectively.

of maspin, PIM-1, and CEA were altered minimally by the 4HR treatment for 24 h (≤5%) (Fig 10C and 10D).

## Effects of 4HR on the expression of angiogenesis-related proteins in HUVECs

HUVECs treated with 4HR for 24 h increased the expression of angiogenesis-related proteins, as follows; angiogenin (29.7% at 16 h), vascular endothelial growth factor A (VEGF-A, 24.1% at 24 h), VEGF-C (19.1% at 24 h), VEGFR2 (15.2% at 8 h), p-VEGFR2 (14% at 16 h), von Willebrand factor (vWF, 17.4% at 16 h), capillary morphogenesis protein 2 (CMG2, 19.7% at 24 h), Fms-related tyrosine kinase 4 (FLT-4, 27.7% at 16 h), lymphatic vessel endothelial hyaluronan receptor 1 (LYVE-1, 28.1% at 24 h), fibrinogen (13.8% at 24 h), HCAM (7.3% at 8 h), ICAM-1 (17.7% at 16 h), and platelet endothelial cell adhesion molecule-1 (PECAM-1, a.k.a. CD31, 19.8% at 8 h). On the other hand, 4HR decreased the expression of hypoxia-inducible factor-1α (HIF-1α, 19.7% at 24 h), fibroblast growth factor-2 (FGF-2, 6.1% at 24 h), platelet-derived growth factor-A (PDGF-A, 5.1% at 16 h), matrix metalloprotease-2 (MMP-2, 10.9% at 16 h), MMP-10 (5% at 16 h), plasminogen (15.6% at 16 h), plasminogen activator inhibitor-1 (PAI-1, 11.7% at 8 h), and VCAM-1 (10.9% at 8 h). The 4HR treatment also increased the expression of endothelin-1 (plays a key role in vascular homeostasis) and kininogen-1 (a constituent of

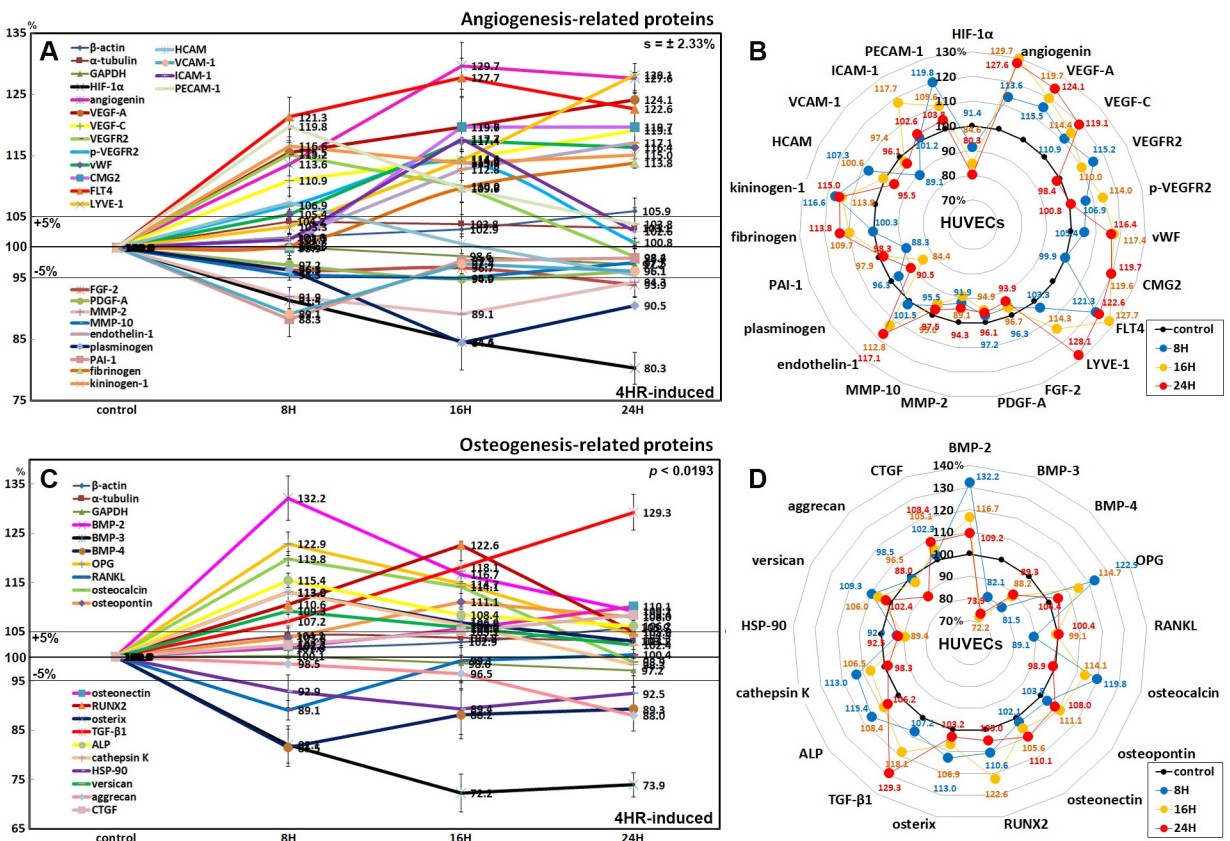

**Fig 11.** Expression of angiogenesis-related proteins (A and B) and osteogenesis-related proteins (C and D) in 4HR-treated HUVECs as determined by IP-HPLC. Line graphs, A and C show protein expression on the same scale (%) versus culture time (8, 16, or 24 h), whereas the star plots (B and D) showed the differential expression levels of the proteins at 8, 16, or 24 h after 4HR administration on the appropriate scales (%). The thick black line, untreated controls (100%); the orange, pink, and red dots show differential protein levels after 4HR administration for 8, 16, or 24 h, respectively.

the blood coagulation system and of the kinin-kallikrein system) by 17.1% and 16.6% at 24 h and 8 h, respectively (Fig 11A and 11B).

## Effects of 4HR on the expression of osteogenesis-related proteins in HUVECs

4HR increased the expression of osteogenesis proteins in HUVECs: BMP-2 (by 38.5% at 8 h), osteoprotegerin (OPG, 22.9% at 8 h), osteocalcin (19.8% at 8 h), osteopontin (11.1% at 16 h), osteonectin (10.1% at 24 h), RUNX2 (22.6% at 16 h), osterix (13% at 8 h), TGF-β1 (29.3% at 24 h), alkaline phosphatase (ALP, 15.4% at 8 h), cathepsin K (a lysosomal cysteine protease involved in bone remodeling and resorption; 13% at 8 h but became similar to control at 24 h), versican (9.3% at 8 h), and CTGF (8.4% at 24 h). On the other hand, 4HR decreased the expression of the osteoclastogenesis-related proteins: receptor activator of nuclear factor kappa-B ligand (RANKL, by 10.9% at 8 h but became similar to control at 24 h) and HSP-90 (osteoclast differentiation factor; 10.6% at 16 h). In contrast, the expression of BMP-3 (a negative regulator for bone density by antagonizing other BMPs), BMP-4 (plays an important role in the onset of endochondral bone formation), and aggrecan (predominant proteoglycan present in cartilage) were downregulated by 27.8%, 18.5%, and 12% at 16 h, 8 h, and 24 h, respectively (Fig 11C and 11D).

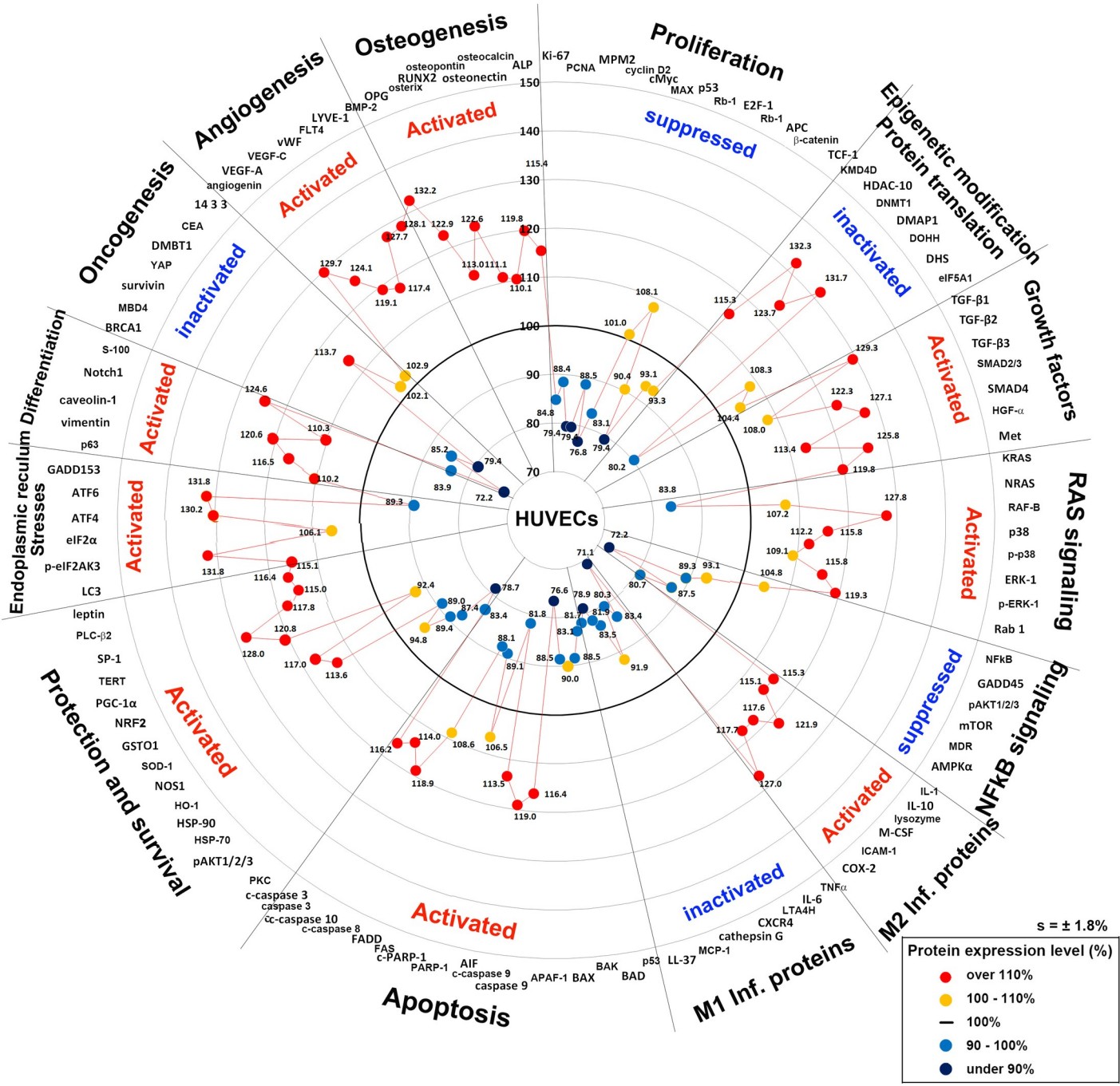

**Fig 12. Star plot of global protein expression in HUVECs treated with 4HR.** The representative proteins (n = 116) selected from 19 major molecular signaling pathways, and their maximum or minimum expression levels (%) were plotted in a circular manner. 4HR activated the growth factors, RAS signaling, M2 macrophage polarization, apoptosis, protection and survival, ER stresses, differentiation, angiogenesis, and osteogenesis, but inactivated proliferation, epigenetic modification-associated DNA transcription, protein translation, NFkB signaling, M1 macrophage polarization, and oncogenesis.

## Global protein expression in 4HR-treated HUVECs

Fig 12 presents the global protein expression changes in the representative proteins (n = 116) of the above 19 different protein signaling pathways as a star plot. The 4HR-treated HUVECs showed similar protein expression changes to the 4HR-treated RAW 264.7 cells. 4HR appeared

to activate cellular proliferation by downregulating the proliferation-related proteins (Ki-67, PCNA, MPM2, and cyclin D2), cMyc/MAX/MAD network proteins (cMyc and MAX), p53/ Rb/E2F signaling (p53 and E2F-1), and Wnt/β-catenin signaling (Wnt1, APC, and TCF-1), to inactivate epigenetic modification by upregulating HDAC10, DNMT1, and DMAP, and to inhibit protein translation by downregulating DOHH and DHS both in HUVECs and RAW 264.7 cells [19].

4HR upregulated the growth factors (TGF-β1, TGF-β3, SMAD2/3, SMAD4, HGFα, and Met) and RAS signaling proteins (RAF-B, p38, p-p38, p-ERK-1, and Rab-1), as well as the expression of cellular protection-related proteins (SOD-1, GSTO1, LC3, PGC-1α, TERT, SP-1, PLC-β2, and leptin), differentiation-related proteins (p63, vimentin, caveolin-1, Notch1, and S-100), and ER stress-related proteins (p-eIF2AK4, eIF2α, ATF4, ATF6, GADD153, and LC3) in HUVECs more than in RAW 264.7 cells. These results coincide with the high growth potential of HUVECs, which can lead to angiogenesis by the increasing endothelial cell differentiation (i.e., 4HR upregulated angiogenin, VEGF-A, VEGF-C, vWF, FLT4, and LYVE-1) and provide osteogenesis-related proteins (BMP-2, OPG, osterix, RUNX2, osteopontin, osteonectin, osteocalcin, and ALP).

4HR can induce anti-inflammatory effects by suppressing NFkB signaling, and downregulating the proteins associated with the M1 macrophage phenotype (TNFα, IL-6, LTA4H, CXCR4, CTLA4, cathepsin G, MCP-1, and LL-37), as well as induce a wound-healing effect by upregulating the proteins associated with the M2 macrophage phenotype (IL-1, IL-10, lysozyme, M-CSF, ICAM-1, and COX-2) in HUVECs. 4HR suppressed p53- and FAS-mediated apoptosis by downregulating p53, BAD, BAK, APAF-1, PARP-1, FAS, and FADD, but induced apoptosis via the PARP-1/AIF pathway associated with some mitochondrial abnormalities by the coincident upregulation of PGC-1α and the downregulation of AMPKα in HUVECs.

The star plot of global protein expression in HUVECs (Fig 12) shows that the 4HR-induced anticancer effect [36, 37, 39–41] was positively influenced by the inactivation of proliferation, epigenetic modification-associated DNA transcription, NFkB signaling, and oncogenesis, and by the activation of apoptosis, ER stresses, cell differentiation including osteogenesis, but negatively influenced by activation of growth factors, RAS signaling, M2 macrophage polarization, cell protection and survival, and angiogenesis, and by the inactivation of M1 macrophage polarization (Fig 13).

On the other hand, the 4HR-induced wound healing effect [18, 20, 21, 35, 42–44] was positively influenced by the activation of growth factors, RAS signaling, M2 macrophage polarization, protection and survival, differentiation including osteogenesis, and angiogenesis, and by the inactivation of NFkB signaling, M1 macrophage polarization and oncogenesis, but it was negatively influenced by the activation of apoptosis and ER stresses, and by the inactivation of proliferation and epigenetic modification-associated DNA transcription (Fig 14).

## Discussion

4HR is a diphenol with a pendant hexane chain, that imparts hydrophobicity to the molecule. 4HR can adhere to some proteins and deactivate them by changing their molecular conformations [45]. In a previous study on 4HR to protein adherence in RAW 264.7 cells [19], MAX, Rb-1, E2F-1, KDM4D, pAKT1/2/3, PKC, p-PKC1α, GADD45, NOS1, TNFα, lysozyme, FGF-2, PDGF-A, FLT4, ERβ, endothelin-1, RANKL, OPG, osteopontin, osteocalcin, and mucin 1 were all found to adhere to 4HR, indicating a direct 4HR interaction to those proteins. The 4HR-induced effects were expected to differ according to the cell types. Therefore, in the present study, HUVECs were used to assess the 4HR-induced effects by IP-HPLC.

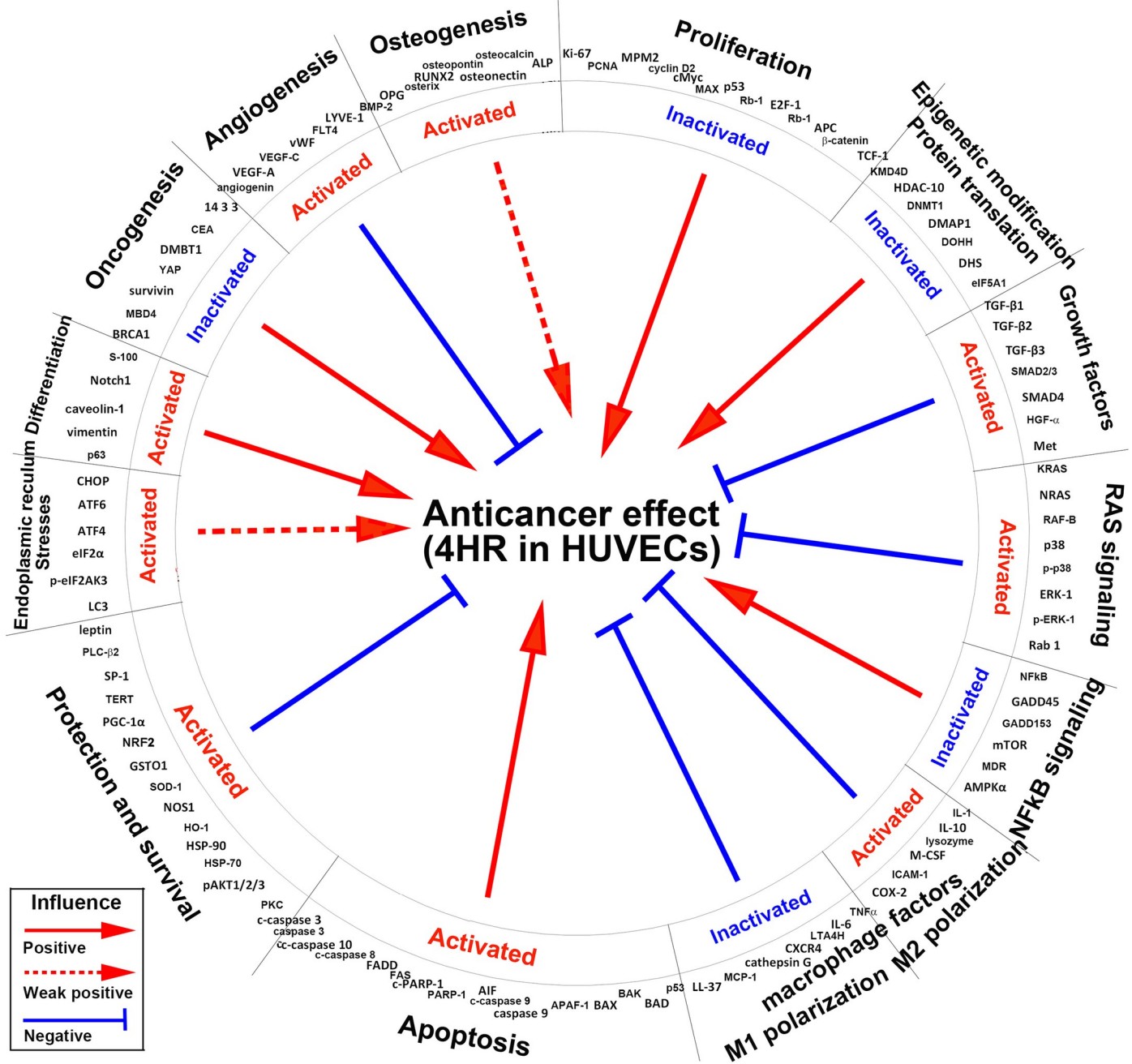

**Fig 13. Global protein signaling pathways contributing for 4HR-induced anticancer effect in HUVECs.** Different protein signaling pathways positively or negatively influenced the anticancer effect. Red arrow line: positive influence; red arrow dot line: weak positive influence; blue inhibition line: negative influence.

A comparison of the 4HR effects between HUVECs and RAW 264.7 cells revealed the 4HR-treated HUVECs to show a downregulation of proliferation-, NFkB signaling-, apoptosis-, and oncogenesis-related proteins similar to 4HR-treated RAW 264.7 cells, while the former showed higher expression of different growth factors, RAS signaling-, M2 macrophage polarization proteins, protection- and survival-, differentiation-, ER stress-, angiogenesis-, and osteogenesis-related proteins than the latter. These results suggest that HUVECs have stronger wound healing properties after a 4HR treatment via the activation of RAS signaling, growth

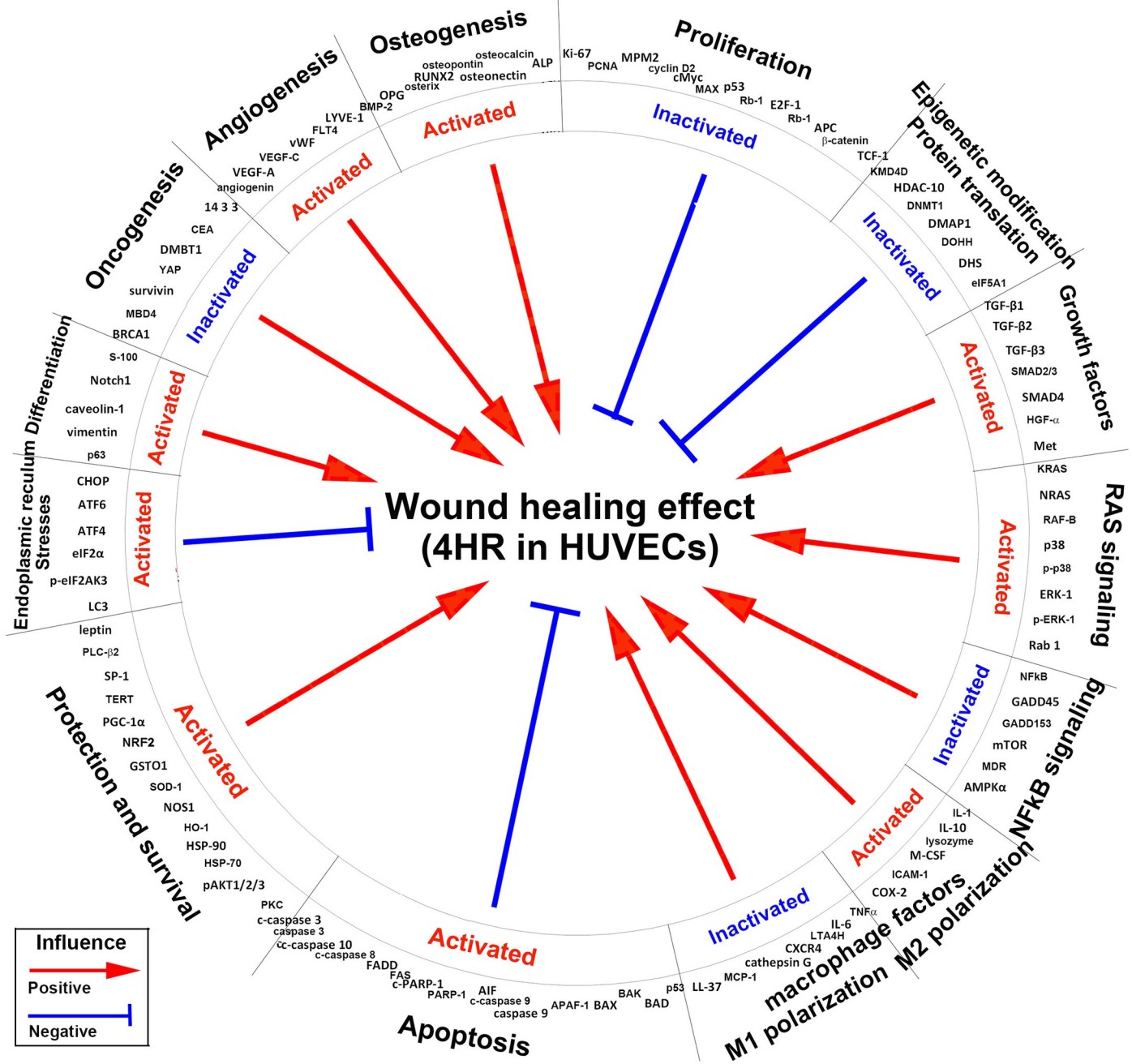

**Fig 14. Global protein signaling pathways contributing to the 4HR-induced wound healing effect in HUVECs.** Different protein signaling pathways positively or negatively influenced the wound healing effect. Red arrow line: positive influence; blue inhibition line: negative influence.

factors, M2 macrophage polarization, cellular protection and survival, and angiogenesis than RAW 264.7 cells. Therefore, 4HR has a similar effect on the protein expression of HUVECs and RAW 264.7 cells, even though their protein expression levels are slightly different. On the other hand, the coincident downregulation of proliferation and upregulation of growth by 4HR may be contradictory in cells. 4HR usually upregulates the reactive growth factors, including TGF-βs, HGFα, IGF-1, and HER1, instead of the major growth factors, including FGF-1, FGF-2, GH, GHRH, PDGF-A, and CTGF. At the same time, it suppresses proliferation

by upregulating different mitosis and cyclin-related proteins. Therefore, the 4HR-induced effects on HUVECs and RAW 264.7 cells are still safe and well balanced by cellular homeostasis.

The proliferation activity of HUVECs was determined by direct cell counting on a culture dish after the 4HR-treatment. The number of HUVECs was gradually decreased by 4HR, resulting in a lower proliferation index depending on the time at 0, 8, 16, and 24 h. These results suggest that the proliferation of HUVECs was inhibited markedly by 4HR. Moreover, the decrease in cell number was closely related to cellular apoptosis, because different apoptosis-executing enzymes including caspase 3, c-caspase 3, c-caspase 8, caspase 9, c-caspase 9, and c-caspase 10 were overexpressed in 4HR-treated HUVECs in IP-HPLC. Western blot also revealed 4HR-induced apoptosis of HUVECs by gradual increases in c-caspase 3 and PARP-1 expression at 8, 16, and 24 h, and by significant increase in AIF at 8 h and a slight increase at 16 and 24 h. c-PARP-1 expression, indicating the inactivation of PARP-1, was reduced at 8 h but recovered gradually to control level at 24 h. These western blot data were similar to the results of IP-HPLC in this study.

4HR reduced Wnt1/β-catenin signaling, and increased the expression of VE-cadherin and E-cadherin. These results are closely associated with the marked downregulation of Wnt1 (a β-catenin activator) and snail (a repressor of the adhesion molecules (cadherins)), and slight upregulation of β-catenin which can stabilize cadherin molecules. On the other hand, higher VE-cadherin expression than E-cadherin was observed in the 4HR-treated HUVECs: 123.6% and 106.2% at 16 h, respectively. The decrease in Wnt1 and β-catenin was coincident with the downregulation of E2F-1 and the suppression of cell proliferation.

The 4HR-treated HUVECs showed overexpression of AIF and caspases despite attenuating p53- and FAS-mediated pro-apoptotic signaling, while the 4HR-treated RAW 264.7 cells showed a marked increase in FAS-mediated apoptosis [19]. AIF was upregulated consistently in HUVECs after the 4HR treatment, and c-PARP-1 was slightly upregulated at 24 h, even though PARP-1 expression was still reduced. Simultaneously, the apoptosis-executing proteins, caspase 3, c-caspase 3, c-caspase 8, caspase 9, c-caspase 9, and c-caspase 10, and PGC-1α, were all upregulated by 4HR. Therefore, 4HR induced alternative apoptosis via PARP-1/AIF signaling associated with mitochondrial damage in HUVECs [46, 47].

Although this study did not determine if 4HR causes mitochondrial membrane damage, 4HR induced abnormal mitochondrial biogenesis by the concomitant upregulation of BID, AIF, and PGC-1α (a master regulator of mitochondrial biogenesis) and the downregulation of AMPKα (a marker of energy consumption). These events resulted in AIF-mediated apoptosis by upregulating caspase 3, 8, 9, which were then activated by the mitochondrial proteins [46–49].

This 4HR-induced cellular apoptosis would be progressive and involved in the alternative activation of NFkB signaling or the compensatory stimulation of TGF-βs production. In the present study, 4HR-treated HUVECs strongly expressed TGF-β1, -β2, and -β3 despite the consistent downregulation of FGF-1, FGF-2, FGF-7, GH, GHRH, PDGF-A, and c-erbB-2 (HER2). The dominant expression of TGF-β1, -β2, and β3 may lead to activation of the SMAD2/3/ SMAD4 pathway, resulting in the transcription of the target genes (e.g., VEGFs and BMPs) and the activations of RAF-B/ERK and p38 signaling [21, 22, 50, 51]. In the present study, these TGF-β signaling cascades were upregulated markedly by 4HR in HUVECs, which increased the expression of RAF-B, SMADs, ERK-1, p38, VEGFs, and BMP-2. Therefore, HUVECs have strong regenerative properties to react with 4HR by upregulating TGF-βs.

The histology examination of the cells spread over the surface of the culture slide dish revealed many small vacuoles in the cytoplasm of 4HR-treated HUVECs compared to the untreated controls. The small vacuoles gradually occupied the entire cytoplasm of HUVECs,

which were strongly positive for LC3 but weakly positive for lysozyme in ICC staining. Therefore, it was assumed that the small vacuoles belong to autophages, resulting from ER stresses induced by 4HR. This assumption was investigated with IP-HPLC, ICC, and western blot analyses. In the IP-HPLC, eIF2AK3, a protein kinase R-like endoplasmic reticulum kinase (PERK), and p-eIF2AK3 were upregulated simultaneously in 8, 16, and 24 h. In contrast, eIF2α was downregulated with overexpression of p-eIF2α in 16 and 24 h. Transcription factors responding to ER stresses, ATF4 and ATF6 were consistently upregulated, but a DNA damage-inducible pro-apoptotic transcription factor, GADD153 was downregulated at 8, 16, and 24 h. These results suggest that eIF2AK3 was active and rapidly phosphorylated into p-eIF2AK3 which subsequently inactivated eIF2α by phosphorylating the alpha subunit of eIF2α, resulting in the repression of global protein synthesis in 4HR-treated cells. The consistent upregulation of ATF4 and ATF6 and the downregulation of GADD153 might rescue 4HR-treated HUVECs from apoptotic damage, as well as the coincident upregulation of LC3 has a protective role of autophagosomes which were found in 4HR-treated cells at 8, 16, and 24 h.

The ICC staining showed a positive reaction of eIF2AK3, eIF2α, ATF4, GADD153, and LC3. The 4HR-treated cells clearly showed the nuclear localization of eIF2AK3, eIF2α, ATF4 and GADD153, as well as the cytoplasmic accumulation of LC3 at 16 and 24 h. Western blot showed strong protein bands of eIF2AK3, ATF4, and LC3 after the 4HR treatment, but the protein bands of eIF2α and GADD153 were decreased or weak. Although both ICC and western blot analysis are unsuitable for accurate quantitative analysis of protein expression, their protein expression trends were similar to the protein expression changes (%) in IP-HPLC performed in the present study. IP-HPLC detected the minor changes in protein expression, and showed the upregulation of ER stress-related proteins. In particular, the expression of phosphor-proteins, p-eIF2AK3 and p-eIF2α was higher than the expression of nonphosphor-proteins. On the other hand, the expression of the ER stress-related proteins usually reached a maximum at 16 h after the 4HR treatment and then tended to decrease at 24 h. Therefore, HUVECs can be recovered partly from the impact of 4HR at 24 h. These results suggest that 4HR induces ER stresses in HUVECs, and produced autophages to induce different cellular functions, including protection, survival, differentiation, and apoptosis.

4HR downregulated the antioxidant proteins (NRF2, SOD-1, SVCT2, and HO-1) and oncogenesis-related proteins (surviving, YAP, CEA, and mucin 1), and upregulated the tumor suppressor proteins (PTEN, BRCA2, NF-1, DMBT1, and ATM) in HUVECs similar to in RAW 264.7 cells. In particular, 4HR suppressed NFkB signaling, but increased the expression of proteins associated with the M2 macrophage phenotype and decreased the expression of protein associated with the M1 macrophage phenotype similarly in both cell types. Therefore, both HUVECs and RAW 264.7 cells may have potent anti-inflammatory and angiogenesis properties after 4HR treatment.

Regarding 4HR-induced angiogenesis effects, HUVECs are probably one of the most appropriate cell types to elucidate the signaling mechanism responsible for 4HR-induced angiogenesis. In the present study, 4HR upregulated many angiogenic proteins (angiogenin, VEGF-A, VEGF-C, VRGFR2, p-VEGFR2, vWF, CMG2, FLT4, and LYVE-1), while downregulated HIF-1α and matrix angiogenesis-related proteins (FGF-2, PDGF-A, MMP-2, plasminogen, and VCAM-1). In addition, to its anti-inflammatory and angiogenesis effects, 4HR consistently upregulated osteogenesis-related proteins (BMP-2, OPG, osteocalcin, osteopontin, osteonectin, RUNX2, osterix, TGF-β1, ALP, versican, and CTGF) in HUVECs. The 4HR-induced osteogenesis-related proteins are usually soluble factors that function in a paracrine manner, indicating that these proteins can stimulate adjacent osteogenic cells to differentiate *in vivo*. Furthermore, the osteogenetic, anti-inflammatory, and angiogenetic effects of 4HR in combination are likely to enhance wound healing and osteogenesis signaling in HUVECs synergistically.

Fig 12 summarizes the global protein expression changes induced by 4HR. The global protein expression changes after 4HR administration were similar in HUVECs and RAW 264.7 cells, but 4HR upregulated some growth factors and some downstream proteins of RAS signaling more in HUVECs than in RAW 264.7 cells. 4HR downregulated antioxidant-related protein expression but upregulated the expression of protection- and survival-, and differentiation-related proteins. 4HR also upregulated TGF-βs/SMADs/VEGFs signaling, RAF-B/ERK and p38 signaling, M2 macrophage polarization, angiogenesis, and osteogenesis, and enhanced caspase activation and subsequent apoptosis.

In addition to comparing the changes in protein expression between 4HR-treated HUVECs and RAW 264.7 cells, this study evaluated the potentials of anticancer and wound healing effects induced by 4HR from the IP-HPLC results. 4HR induced changes in global protein expression and affected the overall protein signaling pathways positively or negatively. The 4HR-induced anticancer effect is already known [36, 37, 39–41] and was simultaneously alleviated by the activation of growth factors, RAS signaling, M2 macrophage polarization, cell protection and survival, and angiogenesis, as well as by the inactivation of M1 macrophage polarization proteins (Fig 13). The overexpression of growth factors (TGF-βs, HGFα, IGF-1, and HER1), cell survival proteins (TERT, SP-1, and PGC-1α), M2 macrophage polarization proteins (IL-10, M-CSF, Pdcd-1/1, and COX-2), and angiogenesis-related proteins (VEGF-A, VEGF-C, and vWF) may be critical to tumor recurrence and metastasis.

The wound-healing effect was alleviated by the inactivation of proliferation, DNA transcription, and protein translation, as well as by apoptosis and ER stresses. Although HUVECs have strong regenerative properties through the higher expression of growth factors, protection, and survival proteins, and angiogenesis-related proteins than RAW 264.7 cells, the suppression of proliferation, DNA transcription, and protein translation may adversely affect HUVECs regeneration, and may eventually lead ER stresses and apoptosis (Fig 14). Despite this, the present study showed consistent trends of 4HR-induced cellular functions exerting anticancer and wound healing procedures both in HUVEC s and RAW 264.7 cells. Therefore, further study may be needed to elucidate the precise molecular cross-talk between different protein signaling pathways of global protein expression.

## Conclusions

4HR-treated HUVECs showed larger increases in the expression of growth factors, RAS signaling proteins, AIF-mediated apoptosis-, protection- and survival-, differentiation-, ER stress-, M2 macrophage polarization- angiogenesis-, and osteogenesis-related proteins than 4HR-treated RAW 274.7 cells, but both cells showed similar trends of decreases in the expression of proliferation-, NFkB signaling- M1 macrophage polarization- and oncogenesis-related proteins, and inactivation of DNA transcription and protein translation. The global protein expression changes induced by 4HR in HUVECs appeared to reveal the anticancer and wound healing effects of 4HR, but the anticancer effect was alleviated by the activation of growth factors, RAS signaling, M2 macrophage polarization proteins, cell protection and survival, and angiogenesis, and by the inactivation of M1 macrophage polarization proteins. In addition, the wound healing effect was alleviated by the inactivation of proliferation, DNA transcription, and protein translation, and by the activation of apoptosis and ER stresses.

## Supporting information

**S1 Data. Mathematical algorithm for IP-HPLC analysis.**
(DOCX)

**S2 Data. Representative chromatography through IP-HPLC analysis.**
(DOCX)

**S1 File.**
(PDF)

## Acknowledgments

We express our gratitude to the late Professor Je Geun Chi who encouraged us to perform IP-HPLC, and to the late Dr. Soo Il Chung who taught us the biological usefulness of IP-HPLC.

## Author Contributions

**Conceptualization:** Yeon Sook Kim, Seong-Gon Kim, Suk Keun Lee.

**Data curation:** Yeon Sook Kim, Dae Won Kim, Suk Keun Lee.

**Formal analysis:** Dae Won Kim, Seong-Gon Kim.

**Investigation:** Yeon Sook Kim, Dae Won Kim, Suk Keun Lee.

**Methodology:** Dae Won Kim.

**Project administration:** Suk Keun Lee.

**Validation:** Seong-Gon Kim, Suk Keun Lee.

**Writing – original draft:** Yeon Sook Kim, Suk Keun Lee.

**Writing – review & editing:** Seong-Gon Kim.

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
