## [Decision Letter · Decision Letter 0]

20 May 2020

PONE-D-20-02990

4-hexylresorcinol-induced protein expression changes in human umbilical cord vein endothelial cells as determined by immunoprecipitation high performance liquid chromatography

PLOS ONE

Dear Dr Lee,

Thank you for submitting your manuscript to PLOS ONE. After careful consideration, we feel that it has merit but does not fully meet PLOS ONE’s publication criteria as it currently stands. Therefore, we invite you to submit a revised version of the manuscript that addresses the points raised during the review process.

Please address the comments in the review, and note that substantial clarification, and additional experiments are required.

We would appreciate receiving your revised manuscript by October 31, 2020. To enhance the reproducibility of your results, we recommend that if applicable you deposit your laboratory protocols in protocols.io, where a protocol can be assigned its own identifier (DOI) such that it can be cited independently in the future. For instructions see: http://journals.plos.org/plosone/s/submission-guidelines#loc-laboratory-protocols

We look forward to receiving your revised manuscript.

Kind regards,

Christina L Addison, Ph.D.

Academic Editor

PLOS ONE

Journal Requirements:

Reviewers' comments:

Reviewer's Responses to Questions

**Comments to the Author**

1. Is the manuscript technically sound, and do the data support the conclusions?

Reviewer #1: Partly

2. Has the statistical analysis been performed appropriately and rigorously? 

Reviewer #1: Yes

3. Have the authors made all data underlying the findings in their manuscript fully available?

Reviewer #1: No

4. Is the manuscript presented in an intelligible fashion and written in standard English?

Reviewer #1: Yes

5. Review Comments to the Author

Reviewer #1: This is an interesting study with potential, but needs some clarification in the methods, as well as additional experiments to be able to support the conclusions. Specific comments are provided in the attached document.

6. PLOS authors have the option to publish the peer review history of their article (what does this mean?). If published, this will include your full peer review and any attached files.

Reviewer #1: No

---

## [Author Response · Author response to Decision Letter 0]

11 Sep 2020

Dear Editor, 

 We have done the major revision of this manuscript by including the new results of cell counting assay, immunohistochemistry, western blot, and additional IP-HPLC (Fig. 1-4 and Fig. 13 and 14). In the text, many sentences were corrected or rewritten according to the reviewer’s comments as follows;

PONE-D-20-02990 - Review 

This study examines the effect of 4-hexylresorcinol (4HR) on human umbilical vein endothelial cells (HUVEC) protein and phospho-protein expression (227 in total) at three time points over a 24-hour period, using IP-HPLC. Protein expression in treated cells was compared to control (untreated) cells and significantly upregulated and downregulated proteins were identified. As these proteins belonged to specific functional groups, this allowed the authors to infer the impact of 4HR in modulators of various physiological processes, including angiogenesis, inflammation, etc. A similar study was previously performed in RAW 264.7 (virus-transformed macrophages), and the purpose of the current study was to investigate whether the effects in HUVEC are similar or different to those found in RAW 264.7 cells. 

Although the study generated a number of interesting observations of potential physiological impact (e.g., upregulation of pro-angiogenic factors, changes indicative of potential growth inhibitory and apoptotic effects, effects on key inflammation mediators, etc.), there are some gaps in the study, which make the manuscript unacceptable at the present form. It is important to add that, if improved, the information gained with this study could be important to our understanding of the effects of 4HR in humans and how this could be exploited for improvements in the food and cosmetic industry (and even biomedicine). 

[Answer]

Recently 4HR has been widely used in the food and cosmetic industry, therefore, the precise cellular effects induced by 4HR is necessary to know 4HR-induced physiological roles or hazard effect. In the previous study, RAW 264.7 cells which are derived from murine monocytes were explored, because they are essential defensing cells which initially engulf or immunologically react with 4HR molecules. And the present study explored HUVECs which play an important role for wound healing procedures. The minute amount of 4HR derived from tooth pastes, food additive, and cosmetic agents may directly affect macrophages (RAW 264.7 cells) and endothelial cells (HUVECs) in vivo. Therefore, we thought this study is helpful to understand the biological behavior of 4HR conveyed through food and cosmetics. 

Below are my specific comments on this manuscript. 

1. The authors mentioned that 4HR is added to foods, tooth paste, and cosmetics (Abstract and Introduction), but it is unclear how this investigation will help (or not) with continuing its application to these industries. In other words, I did not see a clear rationale for this study other than figuring out if the effects they were going to find were different in HUVEC as compared to the ones that were previously found in RAW 264.7 cells. 

[Answer]

In order to make clear the rationale for this experiment, we added a description was added to explain why HUVECs were selected in this study in Introduction section as follow; 

“As the angiogenesis and osteogenesis are important phenomena for tissue organogenesis, it is suggested HUVECs (endothelial cells), counterparts of RAW 264.7 cells (macrophages) in inflammation and wound healing, are also need to be explored for global protein expression changes induced by 4HR.” Line; 76-79

2. How does a 10 micrograms per mL concentration of 4HR compare to what a human being will be exposed to by tooth paste/cosmetic use or food consumption? Is this a physiologically relevant dose? This should have been addressed in the methods section. 

[Answer]

In the previous studies, we explored 4HR effect on cells depending the concentration, 0.1, 1, 5, 10, 20, 50, 100 μg/mL, and then it was found that 10 μg/mL 4HR treatment showed consistent cleat positive results in different cells. Therefore, we choose 10 μg/mL concentration of 4HR treatment in this study. However, regarding the side effect of food additive and cosmetic agents containing 4HR, the absorbed amount of 4HR through gastrointestinal tract, mucosa, and skin might be much smaller than 10 μg/mL. However, there could appear an accumulated effect of 4HR when those agents are used for a longtime. In this study, we want to know unique cellular effect after 4HR treatment within 24 hours. 

A sentence was added to define the reason to select a single concentration of 4HR, 10 μg/mL, as follow;

The concentration of 4HR treatment, 10 μg/mL, was a minimum effective dose determined through the previous studies, which consistently showed positive protein expressions in different cells [12, 13, 25-27]. Lines; 95-97

3. The methods do not explain the IP-HPLC well, why expose the lysate to protein G/A sepharose before exposing to the antibody? Proteins in the lysates would not bind with high affinity to protein G/A, it is the antibodies that do. Perhaps there is a control step first to discard proteins that bind to protein G/A in absence of antibodies? I did not see that control… Steps of the procedure need to be explained better. 

[Answer]

There must be occurred something during manuscript edition, and it was clearly a mistyping, therefore, we correct the sentences as follow in the text; 

“Protein A/G agarose beads column (Amicogen, Jinju, Korea) were separately pre-incubated with 1 μg of 234 different antisera shown in Table 1 in Tris-HCl buffer (pH 7.5, 0.1% Tween 20) at room temperature for 1 h. The supernatant of antibody-incubated column was removed, and then followed by immunoprecipitation procedures.” Lines; 140-143

4. Apart from the analysis of the IP-HPLC results, there is not additional confirmation of the changes in proteins and phopho-proteins, which should have been done for at least a group of them (e.g., the angiogenesis mediators, or the growth factors). 

[Answer]

In this study, several phosphor-proteins were used to compared with the expression of precursor or non-phospor proteins; p-SMAD4, p-JNK, p-ERK-1, p-p38, pPKC1α, pAKT1/2/3, and p-VEGFR2, and at this time, p-eIF2AK3, and p-eIF2α were added in IP-HPLC analysis. Unfortunately, those antibodies were used in the additional experiments of western blot and immunohistochemistry. Therefore, we tried to find some expression changes between phosphor-protein and non-phosphor-proteins in IP-HPLC analysis in this study. 

5. The methods used, including statistics, were appropriate and the results are well described, but the concluding sentences interpreting the results for each of the results subsections (and reiterated in the discussion) are speculative and simplistic. it is impossible to affirm that 4HR modulates any cellular or physiological process (proliferation, apoptosis, angiogenesis, etc.) based only on the upregulation and downregulation of proteins. Further, some of the statements are contradictory. For instance, in lines 201- 202 the authors say “These results suggest 4HR inhibits cell proliferation by downregulation of major proliferation-activating proteins…”. Then in line 286, the authors say “These results indicate that 4HR 2 enhances cellular growth…”. This does not make sense, as you can not inhibit proliferation and enhance cell growth at the same time; as far as I understand it proliferation and cell growth are the same thing. 

[Answer]

The contradictory findings in 4HR-treated HUVECs were compared with the protein expressions in 4HR-treated RAW 264.7 cells in this study. We made a discussion about these contradictory events in discussion section as follow;

“However, the coincident downregulation of proliferation and upregulation of growth by 4HR may be contradictory in cells, and then, 4HR usually upregulated reactive growth factors including TGF-βs, HGFα, IGF-1, and HER1 instead of major growth factors including FGF-1, -2, GH, GHRH, PDGF-A, and CTGF, while suppressed proliferation by upregulating different mitosis and cyclin-related proteins. Therefore, it is believed that 4HR-induced effects on HUVECs and RAW 264.7 cells are still safe and well balanced by cellular homeostasis.” Line; 633-638

And in order to reduce a confusion from the discrepancies in global protein expressions, two diagrams were made to illustrated the overall interactions between signaling pathways (Fig. 13 and 14)

6. To complement the two previous points, the study is lacking three key components: 

a) Confirmatory data for the observed changes for at least some proteins of interest, in the form of Western blots, ELISA, Immunocytochemistry, immunoflouorescence, etc. 

[Answer]

Immunohistochemistry (IHC) and western blot analysis were performed to confirm the expressions of E-cadherin and VE-cadherin for endothelial cell differentiation, the expressions of TGF-β1, caspase 3, and PARP-1 for cellular apoptosis, and the expressions of PERK, eIF2α, ATF4, ATF6, CHOP, and LC3 for ER stresses in 4HR-treated HUVECs. 

IHC and western blot results were added into Result section with Figures 2-4 in the text. 

b) Confirmatory data that a cellular process is actually occurring. The authors mention in lines 591-592 that "some HUVECS conspicuously underwent apoptosis during 24 h of culture in the histological observation…", but did not show this data. Flow cytometry for Annexin V to show that there is indeed apoptosis, or a cell proliferation assay to show that there is inhibition of proliferation; or a cord formation assay to demonstrate pro-angiogenic effects on HUVEC, etc., will be helpful. 

[Answer]

Because highly adherent cells like HUVECs are usually difficult to manipulate for flow-cytometry analysis, we have done direct cell counting on the surfaces of culture dishes to define the proliferation activity if 4HR-treated HUVECs instead of flow-cytometry at this time. The result is illustrated in Fig. 1. 

And to confirm the apoptosis in 4HR-treated HUVECs, the immunohistochemical observation with caspase 3 and PARP-1 antibodies and Western blot analysis with PARP-1 and TGF-β1 were performed and illustrated in Figures 2-4. 

c) Proof that a given protein is actually responsible of a cellular or physiological process. For instance, if the increase in TGF-beta 1 and TGF-beta 3 are confirmed, and the authors are proposing that these are responsible for apoptosis, growth inhibition, increase in VEGF-A expression, etc.; then perhaps they should use a TGF-beta blocking antibody, a small molecule inhibitor of TGF-beta receptor type I/II, or a silencing RNA to block protein expression, and see what happens to apoptosis, VEGF, and/or cell proliferation, etc. The authors could focus on apoptosis for instance, and at least propose a model by which 4HR promotes HUVEC apoptosis. Next, the authors will need to then discuss why inducing endothelial cell apoptosis by these compounds might be detrimental (or not) to humans. Or if the decide to focus on the potential pro-angiogenic effects, then perhaps this compound might have some clinical application in cases when angiogenesis might be beneficial. The way the manuscript is written now it is impossible to see where this research can lead us to, and why it was important to do it. Although this journal does not assess the impact of the research, I believe it is important to highlight the relevance of the research. 

[Answer]

In the serial studies of 4HR effect on HUVECs done in our laboratory, the other paper has already reported the angiogenesis induced via TGF-β signaling assessed by Western blot using TGF-β1 inhibitor, siRNA of TGF-β1, TGF-β receptor inhibitor (Increased Level of Vascular Endothelial Growth Factors by 4-hexylresorcinol is Mediated by Transforming Growth Factor-beta1 and Accelerates Capillary Regeneration in the Burns in Diabetic Animals. International journal of molecular sciences. 2020;21(10)). The present results of TGF-β1 signaling are almost identical with the above paper results. 

7. The manuscript also contains inconsistencies that suggest a limited understanding of the cell system used and the mechanism of action and regulation of the proteins assessed. For instance, E-cadherin upregulation by 4HR treatment is one of the effects reported for Wnt/beta-catenin signalling mediators, but the study was done in HUVEC, which are endothelial cells and therefore expected to express VEcadherin (the vascular type II cadherin) and not E-cadherin (the epithelial type I cadherin). Is this really E-cadherin what the authors identified? If it is indeed E-cadherin, then has it been reported that HUVEC express E-cadherin? Another reason why double-checking with Western blots would have been helpful. The authors proposed that upregulation of E-cadherin by 4HR is due to enhanced binding by betacatenin (< 10% downregulation at 24 h), which resulted in reduced nuclear translocation of the later. What about the Snail downregulation caused by 4HR as well (24.3% at 8 h)? Isn’t Snail a transcription factor that represses cdh1? This component of the equation will have to be taken into consideration for result interpretation if the E-cadherin (or VE-cadherin) results are confirmed with another method. 

[Answer]

In order to correct some inconsistencies of description, the diagrams of Fig. 13 and 14 were added, and many sentences were rewritten to go straight for the experimental goals. 

And IHC staining and Western blot were performed to confirm the expressions of E-cadherin and VE-cadherin at this time. We found both of E-cadherin and VE-cadherin were conspicuously expressed in HUVECs and showed increased expression in 4HR-treated HUVECs. In the immunostainings of E-cadherin and VE-cadherin, there appeared positive reaction in HUVECs after 4HR treatment, and Western blot of E-cadherin and VE-cadherin disclosed slight increase in 4HR-treated HUVECs at 8 and 16 h (Fig. 2-4)

8. Finally, it is recommended for the updated form of the manuscript that the results are presented in a more succinct manner and without any further discussion, and then discussed in the corresponding Discussion section. The Results section of the manuscript was very long and this will make it more reader friendly.

[Answer]

All short comments in Result section were deleted, and these were discussed in Discussion section as the reviewer suggested. And the irrelevant descriptions were all removed. 

Thank you very much for your kind comments.

---

## [Decision Letter · Decision Letter 1]

20 Oct 2020

PONE-D-20-02990R1

4-hexylresorcinol-induced protein expression changes in human umbilical cord vein endothelial cells as determined by immunoprecipitation high performance liquid chromatography

PLOS ONE

Dear Dr. Lee,

Thank you for submitting your manuscript to PLOS ONE. Although the reviewers acknowledge the manuscript is substantially improved, they still feel certain issues remain to be resolved. Thus after careful consideration, we feel that it has merit but does not fully meet PLOS ONE’s publication criteria as it currently stands. Therefore, we invite you to submit a revised version of the manuscript that addresses the points raised during the review process.

Please see reviewer comments and address outstanding issues as highlighted in their report.

We look forward to receiving your revised manuscript.

Kind regards,

Christina L Addison, Ph.D.

Academic Editor

PLOS ONE

Reviewers' comments:

Reviewer's Responses to Questions

**Comments to the Author**

1. If the authors have adequately addressed your comments raised in a previous round of review and you feel that this manuscript is now acceptable for publication, you may indicate that here to bypass the “Comments to the Author” section, enter your conflict of interest statement in the “Confidential to Editor” section, and submit your "Accept" recommendation.

Reviewer #1: (No Response)

2. Is the manuscript technically sound, and do the data support the conclusions?

Reviewer #1: Partly

3. Has the statistical analysis been performed appropriately and rigorously? 

Reviewer #1: Yes

4. Have the authors made all data underlying the findings in their manuscript fully available?

Reviewer #1: Yes

5. Is the manuscript presented in an intelligible fashion and written in standard English?

Reviewer #1: No

6. Review Comments to the Author

Reviewer #1: This was a resubmission of a study that examines the effect of 4-hexylresorcinol (4HR) on human umbilical vein endothelial cells (HUVEC) protein and phospho-protein expression (227 in total) at three time points over a 24-hour period, using IP-HPLC. Protein expression in treated cells was compared to control (untreated) cells and significantly upregulated and downregulated proteins were identified. As these proteins belonged to specific functional groups, this allowed the authors to infer the impact of 4HR in modulators of various physiological processes, including angiogenesis, inflammation, etc. A similar study was previously performed in RAW 264.7 (virus-transformed macrophages), and the purpose of the current study was to investigate whether the effects in HUVEC are similar or different to those found in RAW 264.7 cells.

The study generated a number of interesting observations of potential physiological impact (e.g., upregulation of pro-angiogenic factors, changes indicative of potential growth inhibitory and apoptotic effects, effects on key inflammation mediators, etc.), but there were some gaps in the initial submission, which made the manuscript unacceptable at the time. I specifically raised 8 major points that needed to be addressed. Authors fully or partially addressed points 1, 3, 4, 5, 6, 7 and 8. However, there are still major and minor concerns that need to be addressed.

Major:

1. How does a 10 micrograms per mL concentration of 4HR compare to what a human being will be exposed to by tooth paste/cosmetic use or food consumption? Is this a physiologically relevant dose? This should have been addressed in the methods section. It was specified that it was a concentration which, when previously tested, showed “positive protein expression in different cells”; what does this mean? This need to be clarified in line 96 of the Material and Methods. In addition, the answer in relation to the physiological relevance of this dose is not convincing. It was argued that, although the dose employed in the present study is significantly larger that the dose achieved by cosmetic/food products used by humans, the effect in human can be cumulative. What is the concentration achieved in tissue in humans? I also disagree that a 24 hour treatment with a high dose (acute exposure) is similar to a cumulative effect of lower doses (chronic exposure). If there is some literature the authors could cite to address these persistent concerns, it will greatly enhance the meaning and applicability of their results.

2. The results of the immunocytochemistry of various molecules in Figures 2 and 3 show interesting and potentially relevant changes in subcellular localization of various proteins in HUVEC treated with 4HR, which become more obvious as the time increases. These were not mentioned or discussed. For instance, a change from predominantly nuclear to cytoplasmic seems to be happening for eIF2AK3 and LC3. Is this a valid observation? If this an artefact resulting from the lower magnification at which some images were taken in the 0h control as compared to the other time points? The magnification should be the same in all panels, and subcellular localization differences must be described in results and discussed later on.

3. Although the manuscript was reviewed and it has significantly improved, there are still some parts of the discussion that affirm mechanistic aspects that were not experimentally demonstrated. It is important to revise lines 632-634, 648-653, 654-655 to reflect the fact that results are only suggestive of the mechanism/processes mentioned in those lines but they do not represent a proof that they are actually happening. All this manuscript presents are correlations; the demonstration of the involvement of signalling pathways in specific phenomena requires systematic functional assays that were not performed.

4. Please revise the discussion related to the meaning of Western blot analysis of PARP-1 and Immunocytochemistry of caspase-3 and PARP-1. Increase in these proteins do not necessarily reflect apoptosis. You should have determined their cleaved counterparts, which are the actual markers of apoptosis. In fact, when we have used an antibody that recognizes both the full-length and cleaved form of these proteins, what we observed was a concomitant reduction in the full-length as the levels of the cleaved form went up. The molecular weight of PARP-1 is not shown in the Western blot, but based on the description of the antibody used (Material and Methods section) what is shown by the blots in Fig.4 is an increase in full-length protein.

Minor:

The whole manuscript will benefit from a revision of English grammar and style. Below are some suggestions related to this, missing information, or misleading information.

Abstract

1. Line 30: change “than non-treated” to “as compared to non-treated”.

2. Line 31: Eliminate the word “Whereas” at the beginning of this sentence.

3. Line 34: switch “and had anti-inflammatory…” for “in a manner that suggest potential anti-inflammatory”.

4. Line 36: add the word “mediators” after “ER stresses”.

5. Lines 39-40: Eliminate “that is, HUVECs (endothelial cells) have strong regenerative potential for wound healing, while RAW 264.7 cells (macrophages) play a key role for inflammation”, and adjust punctuation accordingly.

Introduction

Needs a full revision of grammar and style.

Materials and Methods

1. Line 89: concentration of a few growth factors in media is missing.

2. Line 96: not sure what “positive protein expression” means; please explain.

3. Line 109: change the word “immunohistochemical” to “immunocytochemical (ICC)”, as you immunolabelled cells and not tissue. Make sure to change it in the results section and in the figure legends as well.

4. Lines 113 and 121: specify whether antibodies used for PARP-1 and caspase-3 recognize the full-length or the cleaved form; modify the description of corresponding results (Figs. 2 and 4) accordingly

Results

1. Line 256: “condensed” is not an appropriate description of the changes observed; please revise

2. Line 268: eliminate caspase-3 from the result description, as it was not tested by Western blotting. Also eliminate in Figure legend (line 278). Alternatively, if you have a blot for it, then include.

3. Fig. 4: Add MW of proteins to each blot.

4. Line 270-272: confusing sentence; please rewrite.

Discussion

1. Line 674: Not sure what the authors mean by “crosstalk between TGF-beta and SMAD signalling”. There is no crosstalk between TGF-beta and SMAD signalling. SMAD signalling is canonically activated by TGF-beta.

2. Line 703: switch “produce a strong angiogenic effect by upregulating” to “upregulated”.

3. Line 714: please revise grammar in the following sentence “although 4HR more upregulated some growth factors and stimulated downstream of RAS signaling in HUVECs than in RAW 264.7 cells”.

7. PLOS authors have the option to publish the peer review history of their article (what does this mean?). If published, this will include your full peer review and any attached files.

Reviewer #1: No

---

## [Author Response · Author response to Decision Letter 1]

19 Nov 2020

Reviewers' comments:

Reviewer #1: This was a resubmission of a study that examines the effect of 4-hexylresorcinol (4HR) on human umbilical vein endothelial cells (HUVEC) protein and phospho-protein expression (227 in total) at three time points over a 24-hour period, using IP-HPLC. Protein expression in treated cells was compared to control (untreated) cells and significantly upregulated and downregulated proteins were identified. As these proteins belonged to specific functional groups, this allowed the authors to infer the impact of 4HR in modulators of various physiological processes, including angiogenesis, inflammation, etc. A similar study was previously performed in RAW 264.7 (virus-transformed macrophages), and the purpose of the current study was to investigate whether the effects in HUVEC are similar or different to those found in RAW 264.7 cells.

The study generated a number of interesting observations of potential physiological impact (e.g., upregulation of pro-angiogenic factors, changes indicative of potential growth inhibitory and apoptotic effects, effects on key inflammation mediators, etc.), but there were some gaps in the initial submission, which made the manuscript unacceptable at the time. I specifically raised 8 major points that needed to be addressed. Authors fully or partially addressed points 1, 3, 4, 5, 6, 7 and 8. However, there are still major and minor concerns that need to be addressed.

Major:

1. How does a 10 micrograms per mL concentration of 4HR compare to what a human being will be exposed to by tooth paste/cosmetic use or food consumption? Is this a physiologically relevant dose? This should have been addressed in the methods section. It was specified that it was a concentration which, when previously tested, showed “positive protein expression in different cells”; what does this mean? This need to be clarified in line 96 of the Material and Methods. In addition, the answer in relation to the physiological relevance of this dose is not convincing. It was argued that, although the dose employed in the present study is significantly larger that the dose achieved by cosmetic/food products used by humans, the effect in human can be cumulative. What is the concentration achieved in tissue in humans? I also disagree that a 24 hour treatment with a high dose (acute exposure) is similar to a cumulative effect of lower doses (chronic exposure). If there is some literature the authors could cite to address these persistent concerns, it will greatly enhance the meaning and applicability of their results.

[Answer]

Several studies performed using different level of 4HR dose in animals (dogs, rats, mice, and cats) were cited in this text, and explained why the present study selected the dose of 10 μg/mL and performed the culture experiment for 24 h. The general animal experiments for 4HR effects were described in introduction section (line 58-79) and their dose-dependent treatments were described in materials and methods section (line 119-124). 

Line 58-79: Under a two-year gavage study conducted by administering 0, 62.5 or 125 mg/kg (0, 62.5 or 125 μg/g) to groups of 50 F344/N rats and 50 B6C3F1 mice of each sex, five days per week, there was no significant differences in survival and no evidence of carcinogenic activity [9]. The oral LD50 of 4HR was 550 mg/kg body weight in rat [10, 11], 475 mg/kg in Guinea-pig [12], approximately 750 mg/kg in rabbit [12], and 200-1000 mg/kg in mice (subcutaneous injection with 5% 4HR in olive oil; 750-1000 mg/kg, intraperitoneal injection with 5% 4HR in olive oil; 200 mg/kg, with 1% 4HR aqueous emulsion; 300 mg/kg) [11]. The probable oral LD50 of 4-hexylresorcinol in humans has been estimated to be between 500 and 5000 mg/kg [13]. These data indicate 4 HR may have relatively wide range of applicable dose in animals and human, and that the dose used in this study, 10 μg/mL, is within a safe range and might be free of toxic chemical hazard. 

 4HR is excreted via the urine mainly in the form of an ethereal sulfate conjugate [14]. In the animal experiments [15], dogs were given single doses of 1 or 3 g 4HR (equivalent to 100 or 300 mg/kg body weight) as crystals in gelatin capsules or as a solution in olive oil, and its excretion in urine and feces was monitored. After administrating 1 g crystalline compound, 29% of the dose was detected in urine and 67% in feces. When the dose was increased to 3 g, 17% and 73% was excreted in urine and feces, respectively. Urinary excretion was rapid, mainly in the first 6 h, and levels were virtually undetectable 12 h after the lower dose and 24-36 h following the higher dose. When 4HR was administered in olive oil, a dose of 1 g resulted in 17% and 76% was excreted in urine and feces, respectively, while 3 g, 10% and 80% was excreted in urine and feces, respectively. When two men received doses of 1 g 4HR, an average of 18% of the dose was recovered in urine within the first 12 h. Thereafter, the compound was not detected in urine samples. Fecal excretion accounted for 64% of the dose [16]. These results suggest the metabolic degradation of 4HR is vigorous for 6 h and persists until 24 h. Therefore, the present study performed 4HR treatment for 24 h in cell culture experiment. 

Lie 119-124: If 20% of 4HR administered orally was absorbed in different animals, the cellular level dose of 4HR was calculated to be 20 or 60 mg/kg in dog [15], 12 mg/kg in cat [33], 12.5 or 25 mg/kg in rat [9], and 25, 50 or 100 mg/kg in mice [9]. 4HR is commonly employed in 1:1000 solution or glycerite (0.01%, 194 μg/mL) in mouthwashes or pharyngeal antiseptic preparation [34]. Therefore, the dose used in the present study, 10 μg/mL, is safe within the experimental range, and the previous studies used the same dose of 4HR had showed characteristic protein expression in cell culture [20, 21, 35-37]. Cultured cells were harvested with protein lysis buffer (PRO-PREPTM, iNtRON Biotechnology, Daejeon, Korea) in ice, and immediately preserved at -70°C until required. 

2. The results of the immunocytochemistry of various molecules in Figures 2 and 3 show interesting and potentially relevant changes in subcellular localization of various proteins in HUVEC treated with 4HR, which become more obvious as the time increases. These were not mentioned or discussed. For instance, a change from predominantly nuclear to cytoplasmic seems to be happening for eIF2AK3 and LC3. Is this a valid observation? If this an artefact resulting from the lower magnification at which some images were taken in the 0h control as compared to the other time points? The magnification should be the same in all panels, and subcellular localization differences must be described in results and discussed later on.

[Answer]

The immunocytochemcal results of eIF2AK3, eIF2α, ATF4, LC3, lysozyme, and PARP-1 were described in results section (287-293 and 299-311), and also discussed them in discussion section (line 745-759). 

All the figure panels were taken in the same magnification, which were indicated with bar ruler at lower right corner of each figure.

Line 287-293 and 299-311: Caspase 3, an apoptosis executing enzyme, was strongly positive in 4HR-treated cells compared to the untreated control cells, and its immunoreaction was localized at the nuclei (Fig. 2D). PARP-1, a highly error-prone DNA repair pathway microhomolgy-mediated end joining enzyme, was usually positive in the nuclei of untreated control cells, but its immunoreaction was increased gradually in the cytoplasm of 4HR-treated cells at 8, 16, and 24 h (Fig. 2E). Lysozyme, a cationic muiramidase, was diffusely positive in the cytoplasm of untreated control cells, but its immunoreaction was localized at the nuclei of 4HR-treated cells at 8, 16, and 24 h (Fig. 2F).

The immunoreaction of eIF2AK3, a protein kinase R (PKR)-like endoplasmic reticulum kinase (PERK) that can inactivate eIF2α, increased gradually in 4HR-treated cells at 8, 16, and 24 h compared to the untreated control cells. eIF2AK3 was diffusely localized at the cytoplasm and nuclei of cells (Fig. 3A). eIF2α, a regulator of global translation in response to cellular stress, was weakly positive in the untreated control cells, but it increased gradually in 4HR-treated cells at 8, 16, and 24 h (Fig. 3B). ATF4, a master transcription factor during the integrated stress response, was weakly positive in the untreated control cells but became strongly positive and localized at the nuclei of 4HR-treated cells at 8, 16, and 24 h (Fig. 3C).

 GADD153 (CHOP, DDIT3), a DNA damage-inducible transcript 3 pro-apoptotic transcription factor, was weakly positive in the untreated control cells, but its immunoreaction increased slightly in the nuclei of 4HR-treated cells at 8, 16, and 24 h (Fig. 3D). LC3, a biomarker of autophagosomes, was strongly positive in the nuclei but weak in the cytoplasm of the untreated cells. The immunoreaction of LC3 was observed in both the cytoplasm and nuclei of HUVECs, and was higher in 4HR-treated cells at 8, 16, and 24 h, and localized at the cytoplasm of cells but sparse in the nuclei (Fig. 3E). 

Line 745-759: The histology examination of the cells spread over the surface of the culture slide dish revealed many small vacuoles in the cytoplasm of 4HR-treated HUVECs compared to the untreated controls. The small vacuoles gradually occupied the entire cytoplasm of HUVECs, which were strongly positive for LC3 but weakly positive for lysozyme in ICC staining. Therefore, it was assumed that the small vacuoles belong to autophages, resulting from ER stresses induced by 4HR. This assumption was investigated with IP-HPLC, ICC, and western blot analyses. In the IP-HPLC, eIF2AK3, a protein kinase R-like endoplasmic reticulum kinase (PERK), and p-eIF2AK3 were upregulated simultaneously in 8, 16, and 24 h. In contrast, eIF2α was downregulated with overexpression of p-eIF2α in 16 and 24 h. Transcription factors responding to ER stresses, ATF4 and ATF6 were consistently upregulated, but a DNA damage-inducible pro-apoptotic transcription factor, GADD153 was downregulated at 8, 16, and 24 h. These results suggest that eIF2AK3 was active and rapidly phosphorylated into p-eIF2AK3 which subsequently inactivated eIF2α by phosphorylating the alpha subunit of eIF2α, resulting in the repression of global protein synthesis in 4HR-treated cells. The consistent upregulation of ATF4 and ATF6 and the downregulation of GADD153 might rescue 4HR-treated HUVECs from apoptotic damage, as well as the coincident upregulation of LC3 has a protective role of autophagosomes which were found in 4HR-treated cells at 8, 16, and 24 h. 

3. Although the manuscript was reviewed and it has significantly improved, there are still some parts of the discussion that affirm mechanistic aspects that were not experimentally demonstrated. It is important to revise lines 632-634, 648-653, 654-655 to reflect the fact that results are only suggestive of the mechanism/processes mentioned in those lines but they do not represent a proof that they are actually happening. All this manuscript presents are correlations; the demonstration of the involvement of signalling pathways in specific phenomena requires systematic functional assays that were not performed.

[Answer]

The sentences were corrected as indicated by the reviewer’s comment in the manner to describe simple phenomena in the protein expressions rather than to specify their signaling pathways throughout the text. 

4. Please revise the discussion related to the meaning of Western blot analysis of PARP-1 and Immunocytochemistry of caspase-3 and PARP-1. Increase in these proteins do not necessarily reflect apoptosis. You should have determined their cleaved counterparts, which are the actual markers of apoptosis. In fact, when we have used an antibody that recognizes both the full-length and cleaved form of these proteins, what we observed was a concomitant reduction in the full-length as the levels of the cleaved form went up. The molecular weight of PARP-1 is not shown in the Western blot, but based on the description of the antibody used (Material and Methods section) what is shown by the blots in Fig.4 is an increase in full-length protein.

[Answer]

Regarding the results of western blot, the data of c-caspase 3, c-PARP-1, and AIF were added, and the possible evidence of apoptosis was discussed with the elevation of c-PARP-1, a hallmark of apoptosis as an inhibited cleaved form after over-activation of PARP-1, during 8 – 24 h after 4HR treatment, and consistent increase of PARP-1, an effector enzyme for DNA repair and apoptosis in 8, 16, and 24 h (line 724-744). 

The molecular weight of each protein was marked on the panel of western blot. And the antigenic epitopes of caspase 3, c-caspase 3, PARP-1, and PARP-1 were described in materials and methods section. 

Minor:

The whole manuscript will benefit from a revision of English grammar and style. Below are some suggestions related to this, missing information, or misleading information.

Abstract

1. Line 30: change “than non-treated” to “as compared to non-treated”.

[Answer]

It was corrected as the reviewer indicated. 

2. Line 31: Eliminate the word “Whereas” at the beginning of this sentence.

[Answer]

It was deleted as the reviewer indicated.

3. Line 34: switch “and had anti-inflammatory…” for “in a manner that suggest potential anti-inflammatory”.

[Answer]

It was corrected as the reviewer indicated.

4. Line 36: add the word “mediators” after “ER stresses”.

[Answer]

It was corrected as the reviewer indicated.

5. Lines 39-40: Eliminate “that is, HUVECs (endothelial cells) have strong regenerative potential for wound healing, while RAW 264.7 cells (macrophages) play a key role for inflammation”, and adjust punctuation accordingly.

[Answer]

It was deleted as the reviewer indicated, and adjusted its punctuation.

Introduction

Needs a full revision of grammar and style.

[Answer]

The whole text including introduction was re-revised by a professional personnel in Nurisco English Editing Company in Korea. 

Materials and Methods

1. Line 89: concentration of a few growth factors in media is missing.

[Answer]

The concentrations of those growth factors were inserted in the text. 

2. Line 96: not sure what “positive protein expression” means; please explain.

[Answer]

The sentence was rewritten in an appropriate manner to keep from making a confusion. 

Line 122-124: Therefore, the dose used in the present study, 10 μg/mL, is safe within the experimental range, and the previous studies used the same dose of 4HR had showed characteristic protein expression in cell culture [20, 21, 35-37].

3. Line 109: change the word “immunohistochemical” to “immunocytochemical (ICC)”, as you immunolabelled cells and not tissue. Make sure to change it in the results section and in the figure legends as well.

[Answer]

The word “immunohistochemical” was changed into “immunocytochemical (ICC)” in the whole text.

4. Lines 113 and 121: specify whether antibodies used for PARP-1 and caspase-3 recognize the full-length or the cleaved form; modify the description of corresponding results (Figs. 2 and 4) accordingly

[Answer]

The full-length or the cleaved form was described for PARP-1 and c-caspase 3 in the text.

Results

1. Line 256: “condensed” is not an appropriate description of the changes observed; please revise

[Answer]

The word “condensed” was changed into “localized at” in the text.

2. Line 268: eliminate caspase-3 from the result description, as it was not tested by Western blotting. Also eliminate in Figure legend (line 278). Alternatively, if you have a blot for it, then include.

[Answer]

Instead of caspase 3, c-caspase 3 was tested by western blot and its result was described in the text. 

3. Fig. 4: Add MW of proteins to each blot.

[Answer]

Molecular weights of proteins were marked on the western blot figure.

4. Line 270-272: confusing sentence; please rewrite.

[Answer]

Line 270-272 was rewritten as follows; 

Line 323-326: eIF2AK3 (PERK), eIF2α, ATF4, and GADD153, contributing eIF2AK3/eIF2α/ATF4/GADD153 signaling for ER stresses, increased or decreased variably, and the expression of eIF2AK3 and ATF4 increased significantly at 8, 16, and 24 h after 4HR treatment. GADD153 expression changed minimally after the 4HR treatment, while eIF2α expression decreased slightly.

Discussion

1. Line 674: Not sure what the authors mean by “crosstalk between TGF-beta and SMAD signaling”. 

There is no crosstalk between TGF-beta and SMAD signaling. SMAD signaling is canonically activated by TGF-beta.

[Answer]

The description about TGF-β1 and SMAD signaling was rewritten as follows;

Line 736-744: This 4HR-induced cellular apoptosis would be progressive and involved in the alternative activation of NFkB signaling or the compensatory stimulation of TGF-βs production. In the present study, 4HR-treated HUVECs strongly expressed TGF-β1, -β2, and -β3 despite the consistent downregulation of FGF-1, FGF-2, FGF-7, GH, GHRH, PDGF-A, and c-erbB-2 (HER2). The dominant expression of TGF-β1, -β2, and β3 may lead to activation of the SMAD2/3/SMAD4 pathway, resulting in the transcription of the target genes (e.g., VEGFs and BMPs) and the activations of RAF-B/ERK and p38 signaling [21, 22, 50, 51]. In the present study, these TGF-β signaling cascades were upregulated markedly by 4HR in HUVECs, which increased the expression of RAF-B, SMADs, ERK-1, p38, VEGFs, and BMP-2. Therefore, HUVECs have strong regenerative properties to react with 4HR by upregulating TGF-βs. 

2. Line 703: switch “produce a strong angiogenic effect by upregulating” to “upregulated”.

[Answer]

This line was corrected as the reviewer indicated. 

3. Line 714: please revise grammar in the following sentence “although 4HR more upregulated some growth factors and stimulated downstream of RAS signaling in HUVECs than in RAW 264.7 cells”.

[Answer]

This sentence was corrected as follows; 

Line 792-793: but 4HR upregulated some growth factors and some downstream proteins of RAS signaling more in HUVECs than in RAW 264.7 cells.

I am greatly appreciated your kind correction by sharing your precious time. 

Thank you very much, again.

---

## [Editor Report · Decision Letter 2]

2 Dec 2020

4-hexylresorcinol-induced protein expression changes in human umbilical cord vein endothelial cells as determined by immunoprecipitation high-performance liquid chromatography

PONE-D-20-02990R2

Dear Dr. Lee

After reviewing your extensive edits in response to reviews, I am pleased to inform you that your manuscript has been judged scientifically suitable for publication and will be formally accepted for publication once it meets all outstanding technical requirements.

Kind regards,

Christina L Addison, Ph.D.

Academic Editor

PLOS ONE
---

## [Editor Report · Acceptance letter]

7 Dec 2020

PONE-D-20-02990R2 

4-hexylresorcinol-induced protein expression changes in human umbilical cord vein endothelial cells as determined by immunoprecipitation high-performance liquid chromatography 

Dear Dr. Lee:

I'm pleased to inform you that your manuscript has been deemed suitable for publication in PLOS ONE. Congratulations! Your manuscript is now with our production department. 

Kind regards, 

on behalf of

Dr. Christina L Addison 

Academic Editor

PLOS ONE